# SEGA: Variance Reduction via Gradient Sketching

Filip Hanzely[1]    Konstantin Mishchenko[1]    Peter Richtárik[1,2,3]

[1] King Abdullah University of Science and Technology, [2]University of Edinburgh,
[3]Moscow Institute of Physics and Technology

## Abstract

We propose a randomized first order optimization method—SEGA (SkEtched GrAdient)—which progressively throughout its iterations builds a variance-reduced estimate of the gradient from random linear measurements (sketches) of the gradient. In each iteration, SEGA updates the current estimate of the gradient through a sketch-and-project operation using the information provided by the latest sketch, and this is subsequently used to compute an unbiased estimate of the true gradient through a random relaxation procedure. This unbiased estimate is then used to perform a gradient step. Unlike standard subspace descent methods, such as coordinate descent, SEGA can be used for optimization problems with a *non-separable* proximal term. We provide a general convergence analysis and prove linear convergence for strongly convex objectives. In the special case of coordinate sketches, SEGA can be enhanced with various techniques such as *importance sampling*, *minibatching* and *acceleration*, and its rate is up to a small constant factor identical to the best-known rate of coordinate descent.

## 1 Introduction

Consider the optimization problem

$$\min_{x \in \mathbb{R}^n} F(x) \stackrel{\text{def}}{=} f(x) + R(x), \tag{1}$$

where $f : \mathbb{R}^n \to \mathbb{R}$ is smooth and $\mu$–strongly convex, and $R : \mathbb{R}^n \to \mathbb{R} \cup \{+\infty\}$ is a closed convex regularizer. In some applications, $R$ is either the indicator function of a convex set or a sparsity inducing non-smooth penalty such as $\ell_1$-norm. We assume that the *proximal operator* of $R$, defined by $\operatorname{prox}_{\alpha R}(x) \stackrel{\text{def}}{=} \operatorname{argmin}_{y \in \mathbb{R}^n} \left\{ R(y) + \frac{1}{2\alpha} \|y - x\|_{\mathbf{B}}^2 \right\}$, is easily computable (e.g., in closed form). Above we use the weighted Euclidean norm $\|x\|_{\mathbf{B}} \stackrel{\text{def}}{=} \langle x, x \rangle_{\mathbf{B}}^{1/2}$, where $\langle x, y \rangle_{\mathbf{B}} \stackrel{\text{def}}{=} \langle \mathbf{B}x, y \rangle$ is a weighted inner product associated with a positive definite weight matrix $\mathbf{B} \succ 0$. Strong convexity of $f$ is defined with respect to the same product and norm[1].

### 1.1 Gradient sketching

In this paper we design proximal gradient-type methods for solving (1) without assuming that the true gradient of $f$ is available. Instead, we assume that an oracle provides a *random linear transformation (i.e., a sketch) of the gradient*, which is the information available to drive the iterative

process. In particular, given a fixed distribution $\mathcal{D}$ over matrices $\mathbf{S} \in \mathbb{R}^{n \times b}$ ($b \geq 1$ can but does not need to be fixed), and a query point $x \in \mathbb{R}^n$, our oracle provides us the random linear transformation of the gradient given by

$$\zeta(\mathbf{S}, x) \stackrel{\text{def}}{=} \mathbf{S}^\top \nabla f(x) \in \mathbb{R}^b, \qquad \mathbf{S} \sim \mathcal{D}. \tag{2}$$

Information of this type is available/used in a variety of scenarios. For instance, randomized coordinate descent (CD) methods use oracle (2) with $\mathcal{D}$ corresponding to a distribution over standard basis vectors. Minibatch/parallel variants of CD methods utilize oracle (2) with $\mathcal{D}$ corresponding to a distribution over random column submatrices of the identity matrix. If one is prepared to use difference of function values to approximate directional derivatives, one can apply our oracle model to zeroth-order optimization [8]. Indeed, the directional derivative of $f$ in a random direction $\mathbf{S} = s \in \mathbb{R}^{n \times 1}$ can be approximated by $\zeta(s, x) \approx \frac{1}{\epsilon}(f(x + \epsilon s) - f(x))$, where $\epsilon > 0$ is sufficiently small.

We now illustrate this concept using two examples.

**Example 1.1** (Sketches)**.** (i) Coordinate sketch. *Let $\mathcal{D}$ be the uniform distribution over standard unit basis vectors $e_1, e_2, \ldots, e_n$ of $\mathbb{R}^n$. Then $\zeta(e_i, x) = e_i^\top \nabla f(x)$, i.e., the $i^{th}$ partial derivative of $f$ at $x$.* (ii) Gaussian sketch. *Let $\mathcal{D}$ be the standard Gaussian distribution in $\mathbb{R}^n$. Then for $s \sim \mathcal{D}$ we have $\zeta(s, x) = s^\top \nabla f(x)$, i.e., the directional derivative of $f$ at $x$ in direction $s$.*

## 1.2 Related work

In the last decade, stochastic gradient-type methods for solving problem (1) have received unprecedented attention by theoreticians and practitioners alike. Specific examples of such methods are stochastic gradient descent (SGD) [43], variance-reduced variants of SGD such as SAG [44], SAGA [10], SVRG [22], and their accelerated counterparts [26, 1]. While these methods are specifically designed for objectives formulated as an expectation or a finite sum, we do not assume such a structure. Moreover, these methods utilize a fundamentally different stochastic gradient information: they have access to an unbiased gradient estimator. In contrast, we do not assume that (2) is an unbiased estimator of $\nabla f(x)$. In fact, $\zeta(\mathbf{S}, x) \in \mathbb{R}^b$ and $\nabla f(x) \in \mathbb{R}^n$ do not even necessarily belong to the same space. Therefore, our algorithms and results are complementary to the above line of research.

While the gradient sketch $\zeta(\mathbf{S}, x)$ does not immediatey lead to an unbiased estimator of the gradient, SEGA uses the information provided in the sketch to *construct* an unbiased estimator of the gradient via a *sketch-and-project* process. Sketch-and-project iterations were introduced in [15] in the contex of linear feasibility problems. A dual view uncovering a direct relationship with stochastic subspace ascent methods was developed in [16]. The latest and most in-depth treatment of sketch-and-project for linear feasibility is based on the idea of stochastic reformulations [42]. Sketch-and-project can be combined with Polyak [29, 28] and Nesterov momentum [14, 47], extended to convex feasibility problems [30], matrix inversion [18, 17, 14], and empirical risk minimization [13, 19].

The line of work most closely related to our setup is that on randomized coordinate/subspace descent methods [34, 16]. Indeed, the information available to these methods is compatible with our oracle for specific distributions $\mathcal{D}$. However, the main disadvantage of these methods is that they can not handle non-separable regularizers $R$. In contrast, the algorithm we propose—SEGA—works for any regularizer $R$. In particular, SEGA can handle non-separable constraints even with coordinate sketches, which is out of range of current CD methods. Hence, our work could be understood as extending the reach of coordinate and subspace descent methods from separable to arbitrary regularizers, which allows for a plethora of new applications. Our method is able to work with an arbitrary regularizer due to its ability to *build an unbiased variance-reduced estimate of the gradient* of $f$ throughout the iterative process from the random sketches provided by the oracle. Moreover, and unlike coordinate descent, SEGA allows for general sketches from essentially any distribution $\mathcal{D}$.

Another stream of work on designing gradient-type methods without assuming perfect access to the gradient is represented by the *inexact gradient descent* methods [9, 11, 45]. However, these methods deal with deterministic estimates of the gradient and are not based on linear transformations of the gradient. Therefore, this second line of research is also significantly different from what we do here.

## 1.3   Outline

We describe SEGA in Section 2. Convergence results for general sketches are described in Section 3. Refined results for coordinate sketches are presented in Section 4, where we also describe and analyze an accelerated variant of SEGA. Experimental results can be found in Section 5. Conclusions are drawn and potential extensions outlined in Appendix A. Proofs of the main results can be found in Appendices B and C. An aggressive *subspace* variant of SEGA is described and analyzed in Appendix D. A simplified analysis of SEGA in the case of coordinate sketches and for $R \equiv 0$ is developed in Appendix E (under standard assumptions as in the main paper) and F (under alternative assumptions). Extra experiments for additional insights are included in Appendix G.

**Notation.**   We introduce notation where needed. We also provide a notation table in Appendix H.

## 2   The SEGA Algorithm

In this section we introduce a learning process for estimating the gradient from the sketched information provided by (2); this will be used as a subroutine of SEGA.

Let $x^k$ be the current iterate, and let $h^k$ be the current estimate of the gradient of $f$. The oracle queried, and we receive new information in the form of the sketched gradient (2). Then, we would like to update $h^k$ based on the new information. We do this using a *sketch-and-project* process [15, 16, 42]: we set $h^{k+1}$ to be the closest vector to $h^k$ (in a certain Euclidean norm) satisfying (2):

$$h^{k+1} = \arg\min_{h \in \mathbb{R}^n} \|h - h^k\|_{\mathbf{B}}^2 \qquad \text{subject to} \quad \mathbf{S}_k^\top h = \mathbf{S}_k^\top \nabla f(x^k). \qquad (3)$$

The closed-form solution of (3) is

$$h^{k+1} = h^k - \mathbf{B}^{-1}\mathbf{Z}_k(h^k - \nabla f(x^k)) = (\mathbf{I} - \mathbf{B}^{-1}\mathbf{Z}_k)h^k + \mathbf{B}^{-1}\mathbf{Z}_k\nabla f(x^k), \qquad (4)$$

where $\mathbf{Z}_k \stackrel{\text{def}}{=} \mathbf{S}_k \left(\mathbf{S}_k^\top \mathbf{B}^{-1}\mathbf{S}_k\right)^\dagger \mathbf{S}_k^\top$. Notice that $h^{k+1}$ is a *biased* estimator of $\nabla f(x^k)$. In order to obtain an unbiased gradient estimator, we introduce a random variable[2] $\theta_k = \theta(\mathbf{S}_k)$ for which

$$\mathbb{E}_{\mathcal{D}}\left[\theta_k \mathbf{Z}_k\right] = \mathbf{B}. \qquad (5)$$

If $\theta_k$ satisfies (5), it is straightforward to see that the random vector

$$g^k \stackrel{\text{def}}{=} (1 - \theta_k)h^k + \theta_k h^{k+1} \stackrel{(4)}{=} h^k + \theta_k \mathbf{B}^{-1}\mathbf{Z}_k(\nabla f(x^k) - h^k) \qquad (6)$$

is an *unbiased estimator* of the gradient:

$$\mathbb{E}_{\mathcal{D}}\left[g^k\right] \stackrel{(5)+(6)}{=} \nabla f(x^k). \qquad (7)$$

Finally, we use $g^k$ instead of the true gradient, and perform a proximal step with respect to $R$. This leads to a new optimization method, which we call *SkEtched GrAdient Method (SEGA)* and describe in Algorithm 1. We stress again that the method does not need the access to the full gradient.

**Algorithm 1:** SEGA: SkEtched GrAdient Method

---

1 **Initialize:** $x^0, h^0 \in \mathbb{R}^n$; $\mathbf{B} \succ 0$; distribution $\mathcal{D}$;
         stepsize $\alpha > 0$
2 **for** $k = 1, 2, \dots$ **do**
3     Sample $\mathbf{S}_k \sim \mathcal{D}$
4     $g^k = h^k + \theta_k \mathbf{B}^{-1} \mathbf{Z}_k (\nabla f(x^k) - h^k)$
5     $x^{k+1} = \text{prox}_{\alpha R}(x^k - \alpha g^k)$
6     $h^{k+1} = h^k + \mathbf{B}^{-1} \mathbf{Z}_k (\nabla f(x^k) - h^k)$

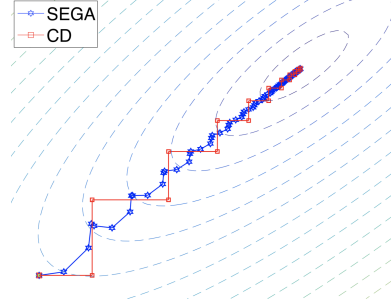

Figure 1: Iterates of SEGA and CD

## 2.1 SEGA as a variance-reduced method

As we shall show, both $h^k$ and $g^k$ become better at approximating $\nabla f(x^k)$ as the iterates $x^k$ approach the optimum. Hence, the variance of $g^k$ as an estimator of the gradient tends to zero, which means that SEGA is a *variance-reduced* algorithm. The structure of SEGA is inspired by the JackSketch algorithm introduced in [19]. However, as JackSketch is aimed at solving a finite-sum optimization problem with many components, it does not make much sense to apply it to (1). Indeed, when applied to (1) (with $R = 0$, since JackSketch was analyzed for smooth optimization only), JackSketch reduces to gradient descent. While JackSketch performs *Jacobian* sketching (i.e., multiplying the Jacobian by a random matrix from the right, effectively sampling a subset of the gradients forming the finite sum), SEGA multiplies the Jacobian by a random matrix from the left. In doing so, SEGA becomes oblivious to the finite-sum structure and transforms into the gradient sketching mechanism described in (2).

## 2.2 SEGA versus coordinate descent

We now illustrate the above general setup on the simple example when $\mathcal{D}$ corresponds to a distribution over standard unit basis vectors in $\mathbb{R}^n$.

**Example 2.1.** *Let* $\mathbf{B} = \text{Diag}(b_1, \dots, b_n) \succ 0$ *and let* $\mathcal{D}$ *be defined as follows. We choose* $\mathbf{S}_k = e_i$ *with probability* $p_i > 0$*, where* $e_1, e_2, \dots, e_n$ *are the unit basis vectors in* $\mathbb{R}^n$*. Then*

$$h^{k+1} \overset{(4)}{=} h^k + e_i^\top (\nabla f(x^k) - h^k) e_i, \tag{8}$$

*which can equivalently be written as* $h_i^{k+1} = e_i^\top \nabla f(x^k)$ *and* $h_j^{k+1} = h_j^k$ *for* $j \neq i$*. Note that* $h^{k+1}$ *does not depend on* $\mathbf{B}$*. If we choose* $\theta_k = \theta(\mathbf{S}_k) = 1/p_i$*, then*

$$\mathbb{E}_{\mathcal{D}} [\theta_k \mathbf{Z}_k] = \sum_{i=1}^n p_i \tfrac{1}{p_i} e_i (e_i^\top \mathbf{B}^{-1} e_i)^{-1} e_i^\top = \sum_{i=1}^n \tfrac{e_i e_i^\top}{1/b_i} = \mathbf{B}$$

*which means that* $\theta_k$ *is a bias-correcting random variable. We then get*

$$g^k \overset{(6)}{=} h^k + \tfrac{1}{p_i} e_i^\top (\nabla f(x^k) - h^k) e_i. \tag{9}$$

In the setup of Example 2.1, both SEGA and CD obtain new gradient information in the form of a random partial derivative of $f$. However, the two methods perform a different update: (i) SEGA allows for arbitrary proximal term, CD allows for separable one only [46, 27, 12]; (ii) While SEGA updates all coordinates in every iteration, CD updates a single coordinate only; (iii) If we force $h^k = 0$ in SEGA and use coordinate sketches, the method transforms into CD.

Based on the above observations, we conclude that SEGA can be applied in more general settings for the price of potentially more expensive iterations[3]. For intuition-building illustration of how SEGA

works, Figure 1 shows the evolution of iterates of both SEGA and CD applied to minimizing a simple quadratic function in 2 dimensions. For more figures of this type, including the composite case where CD does not work, see Appendix G.1.

In Section 4 we show that SEGA enjoys, up to a small constant factor, the same theoretical iteration complexity as CD. This remains true when comparing state-of-the-art variants of CD with importance sampling, parallelism/mini-batching and acceleration with the corresponding variants of SEGA.

**Remark 2.2.** *Nontrivial sketches* $\mathbf{S}$ *and metric* $\mathbf{B}$ *might, in some applications, bring a substantial speedup against the baseline choices mentioned in Example 2.1. Appendix D provides one example: there are problems where the gradient of $f$ always lies in a particular $d$-dimensional subspace of $\mathbb{R}^n$. In such a case, suitable choice of $\mathbf{S}$ and $\mathbf{B}$ leads to $\mathcal{O}\left(\frac{n}{d}\right)$–times faster convergence compared to the setup of Example 2.1. In Section 5.3 we numerically demonstrate this claim.*

# 3 Convergence of SEGA for General Sketches

In this section we state a linear convergence result for SEGA (Algorithm 1) for general sketch distributions $\mathcal{D}$ under smoothness and strong convexity assumptions.

## 3.1 Smoothness assumptions

We will use the following general version of smoothness.

**Assumption 3.1** ($\mathbf{Q}$-smoothness). *Function $f$ is $\mathbf{Q}$-smooth with respect to $\mathbf{B}$, where $\mathbf{Q} \succ 0$ and $\mathbf{B} \succ 0$. That is, for all $x, y$, the following inequality is satisfied:*

$$f(x) - f(y) - \langle \nabla f(y), x - y \rangle_{\mathbf{B}} \geq \tfrac{1}{2}\|\nabla f(x) - \nabla f(y)\|_{\mathbf{Q}}^2, \tag{10}$$

Assumption 3.1 is not standard in the literature. However, as Lemma B.1 states, in the special case of $\mathbf{B} = \mathbf{I}$ and $\mathbf{Q} = \mathbf{M}^{-1}$, it reduces to $\mathbf{M}$-smoothness (see Assumption 3.2), which is a common assumption in modern analysis of CD methods.

**Assumption 3.2** ($\mathbf{M}$-smoothness). *Function $f$ is $\mathbf{M}$-smooth for some matrix $\mathbf{M} \succ 0$. That is, for all $x, y$, the following inequality is satisfied:*

$$f(x) \leq f(y) + \langle \nabla f(y), x - y \rangle + \tfrac{1}{2}\|x - y\|_{\mathbf{M}}^2. \tag{11}$$

Assumption 3.2 is fairly standard in the CD literature. It appears naturally in various application such as empirical risk minimization with linear predictors and is a baseline in the development of minibatch CD methods [41, 38, 36, 39]. We will adopt this notion in Section 4, when comparing SEGA to coordinate descent. Until then, let us consider the more general Assumption 3.1.

## 3.2 Main result

Now we present one of the key theorems of the paper, stating a linear convergence of SEGA.

**Theorem 3.3.** *Assume that $f$ is $\mathbf{Q}$–smooth with respect to $\mathbf{B}$, and $\mu$–strongly convex. Fix $x^0, h^0 \in \mathrm{dom}(F)$ and let $x^k, h^k$ be the random iterates produced by SEGA. Choose stepsize $\alpha > 0$ and Lyapunov parameter $\sigma > 0$ so that*

$$\alpha\left(2(\mathbf{C} - \mathbf{B}) + \sigma\mu\mathbf{B}\right) \leq \sigma\mathbb{E}_{\mathcal{D}}\left[\mathbf{Z}\right], \qquad \alpha\mathbf{C} \leq \tfrac{1}{2}\left(\mathbf{Q} - \sigma\mathbb{E}_{\mathcal{D}}\left[\mathbf{Z}\right]\right), \tag{12}$$

*where $\mathbf{C} \overset{def}{=} \mathbb{E}_{\mathcal{D}}\left[\theta_k^2 \mathbf{Z}_k\right]$. Then $\mathbb{E}\left[\Phi^k\right] \leq (1 - \alpha\mu)^k \Phi^0$ for Lyapunov function $\Phi^k \overset{def}{=} \|x^k - x^*\|_{\mathbf{B}}^2 + \sigma\alpha\|h^k - \nabla f(x^*)\|_{\mathbf{B}}^2$, where $x^*$ is a solution of (1).*

| | CD | SEGA |
|---|---|---|
| Nonaccelerated method importance sampling, $b = 1$ | $\frac{\mathrm{Trace}(\mathbf{M})}{\mu} \log \frac{1}{\epsilon}$ [34] | $8.55 \cdot \frac{\mathrm{Trace}(\mathbf{M})}{\mu} \log \frac{1}{\epsilon}$ |
| Nonaccelerated method arbitrary sampling | $\left( \max_i \frac{v_i}{p_i \mu} \right) \log \frac{1}{\epsilon}$ [41] | $8.55 \cdot \left( \max_i \frac{v_i}{p_i \mu} \right) \log \frac{1}{\epsilon}$ |
| Accelerated method importance sampling, $b = 1$ | $1.62 \cdot \frac{\sum_i \sqrt{\mathbf{M}_{ii}}}{\sqrt{\mu}} \log \frac{1}{\epsilon}$ [3] | $9.8 \cdot \frac{\sum_i \sqrt{\mathbf{M}_{ii}}}{\sqrt{\mu}} \log \frac{1}{\epsilon}$ |
| Accelerated method arbitrary sampling | $1.62 \cdot \sqrt{\max_i \frac{v_i}{p_i^2 \mu}} \log \frac{1}{\epsilon}$ [20] | $9.8 \cdot \sqrt{\max_i \frac{v_i}{p_i^2 \mu}} \log \frac{1}{\epsilon}$ |

Table 1: Complexity results for coordinate descent (CD) and our sketched gradient method (SEGA), specialized to coordinate sketching, for $\mathbf{M}$–smooth and $\mu$–strongly convex functions.

Note that $\Phi^k \to 0$ implies $h^k \to \nabla f(x^*)$. Therefore SEGA is *variance reduced*, in contrast to CD in the non-separable proximal setup, which does not converge to the solution. If $\sigma$ is small enough so that $\mathbf{Q} - \sigma \mathbb{E}_{\mathcal{D}}[\mathbf{Z}] \succ 0$, one can always choose stepsize $\alpha$ satisfying

$$\alpha \leq \min \left\{ \frac{\lambda_{\min}(\mathbb{E}_{\mathcal{D}}[\mathbf{Z}])}{\lambda_{\max}(2\sigma^{-1}(\mathbf{C}-\mathbf{B})+\mu\mathbf{B})}, \frac{\lambda_{\min}(\mathbf{Q}-\sigma\mathbb{E}_{\mathcal{D}}[\mathbf{Z}])}{2\lambda_{\max}(\mathbf{C})} \right\} \tag{13}$$

and inequalities (12) will hold. Therefore, we get the next corollary.

**Corollary 3.4.** *If* $\sigma < \frac{\lambda_{\min}(\mathbf{Q})}{\lambda_{\max}(\mathbb{E}_{\mathcal{D}}[\mathbf{Z}])}$, $\alpha$ *satisfies* (13) *and* $k \geq \frac{1}{\alpha\mu} \log \frac{\Phi^0}{\epsilon}$, *then* $\mathbb{E}\left[\|x^k - x^*\|_{\mathbf{B}}^2\right] \leq \epsilon$.

As Theorem 3.3 is rather general, we also provide a simplified version thereof, complete with a simplified analysis (Theorem E.1 in Appendix E). In the simplified version we remove the proximal setting (i.e., we set $R = 0$), assume $L$–smoothness[4], and only consider coordinate sketches with uniform probabilities. The result is provided as Corollary 3.5.

**Corollary 3.5.** *Let* $\mathbf{B} = \mathbf{I}$ *and choose* $\mathcal{D}$ *to be the uniform distribution over unit basis vectors in* $\mathbb{R}^n$. *If the stepsize satisfies* $0 < \alpha \leq \min\{(1 - L\sigma/n)/(2Ln), n^{-1}(\mu + 2(n-1)/\sigma)^{-1}\}$, *then* $\mathbb{E}_{\mathcal{D}}\left[\Phi^{k+1}\right] \leq (1 - \alpha\mu)\Phi^k$, *and therefore the iteration complexity is* $\tilde{\mathcal{O}}(nL/\mu)$.

**Remark 3.6.** *In the fully general case, one might choose* $\alpha$ *to be bigger than bound* (13), *which depends on eigen properties of* $\mathbb{E}_{\mathcal{D}}[\mathbf{Z}], \mathbf{C}, \mathbf{Q}, \mathbf{B}$, *leading to a better overall complexity. However, in the simple case with* $\mathbf{B} = \mathbf{I}$, $\mathbf{Q} = \mathbf{I}$ *and* $\mathbf{S}_k = e_{i_k}$ *with uniform probabilities, bound* (13) *is tight.*

# 4 Convergence of SEGA for Coordinate Sketches

In this section we compare SEGA with coordinate descent. We demonstrate that, specialized to a particular choice of the distribution $\mathcal{D}$ (where $\mathbf{S}$ is a random column submatrix of the identity matrix), which makes SEGA use the same random gradient information as that used in modern randomized CD methods, SEGA attains, up to a small constant factor, the same convergence rate as CD methods.

Firstly, in Section 4.2 we develop SEGA with in a general setup known as *arbitrary sampling* [41, 40, 37, 38, 6] (Theorem 4.2). Then, in Section 4.3 we develop an *accelerated variant of SEGA* (see Theorem C.5) for arbitrary sampling as well. Lastly, Corollary 4.3 and Corollary 4.4 provide us with *importance sampling* for both nonaccelerated and accelerated method, which matches up to a constant factor cutting-edge CD rates [41, 3] under the same oracle and assumptions[5]. Table 1 summarizes the results of this section. We provide all proofs for this section in Appendix C.

We now describe the setup and technical assumptions for this section. In order to facilitate a direct comparison with CD (which does not work with non-separable regularizer $R$), for simplicity we consider problem (1) in the simplified setting with $R \equiv 0$. Further, function $f$ is assumed to be M–smooth (Assumption 3.2) and $\mu$–strongly convex.

## 4.1 Defining $\mathcal{D}$: samplings

In order to draw a direct comparison with general variants of CD methods (i.e., with those analyzed in the *arbitrary sampling* paradigm), we consider sketches in (3) that are column submatrices of the identity matrix: $\mathbf{S} = \mathbf{I}_S$, where $S$ is a random subset (aka *sampling*) of $[n] \stackrel{\text{def}}{=} \{1, 2, \ldots, n\}$. Note that the columns of $\mathbf{I}_S$ are the standard basis vectors $e_i$ for $i \in S$ and hence $\text{Range}(\mathbf{S}) = \text{Range}(e_i : i \in S)$. So, distribution $\mathcal{D}$ from which we draw matrices is uniquely determined by the distribution of sampling $S$. Given a sampling $S$, define $p = (p_1, \ldots, p_n) \in \mathbb{R}^n$ to be the vector satisfying $p_i = \mathbb{P}(e_i \in \text{Range}(\mathbf{S})) = \mathbb{P}(i \in S)$, and $\mathbf{P}$ to be the matrix for which $\mathbf{P}_{ij} = \mathbb{P}(\{i, j\} \subseteq S)$. Note that $p$ and $\mathbf{P}$ are the *probability vector* and *probability matrix* of sampling $S$, respectively [38]. We assume throughout the paper that $S$ is proper, i.e., we assume that $p_i > 0$ for all $i$. State-of-the-art minibatch CD methods (including the ones we compare against [41, 20]) utilize large stepsizes related to the so-called ESO *Expected Separable Overapproximation (ESO)* [38] parameters $v = (v_1, \ldots, v_n)$. ESO parameters play a key role in SEGA as well, and are defined next.

**Assumption 4.1** (ESO). *There exists a vector $v$ satisfying the following inequality*

$$\mathbf{P} \circ \mathbf{M} \preceq \text{Diag}(p)\text{Diag}(v), \tag{14}$$

*where $\circ$ denotes the Hadamard (i.e., element-wise) product of matrices.*

In case of single coordinate sketches, parameters $v$ are equal to coordinate-wise smoothness constants of $f$. An extensive study on how to choose them in general was performed in [38]. For notational brevity, let us set $\hat{\mathbf{P}} \stackrel{\text{def}}{=} \text{Diag}(p)$ and $\hat{\mathbf{V}} \stackrel{\text{def}}{=} \text{Diag}(v)$ throughout this section.

## 4.2 Non-accelerated method

We now state the convergence rate of (non-accelerated) SEGA for coordinate sketches with *arbitrary sampling* of subsets of coordinates. The corresponding CD method was developed in [41].

**Theorem 4.2.** *Assume that $f$ is M–smooth and $\mu$–strongly convex. Denote $\Psi^k \stackrel{\text{def}}{=} f(x^k) - f(x^*) + \sigma\|h^k\|_{\hat{\mathbf{P}}^{-1}}^2$. Choose $\alpha, \sigma > 0$ such that*

$$\sigma\mathbf{I} - \alpha^2(\hat{\mathbf{V}}\hat{\mathbf{P}}^{-1} - \mathbf{M}) \succeq \gamma\mu\sigma\hat{\mathbf{P}}^{-1}, \tag{15}$$

*where $\gamma \stackrel{\text{def}}{=} \alpha - \alpha^2 \max_i\{\frac{v_i}{p_i}\} - \sigma$. Then the iterates of SEGA satisfy $\mathbb{E}\left[\Psi^k\right] \leq (1 - \gamma\mu)^k\Psi^0$.*

We now give an importance sampling result for a coordinate version of SEGA. We recover, up to a constant factor, the same convergence rate as standard CD [34]. The probabilities we chose are optimal in our analysis and are proportional to the diagonal elements of matrix $\mathbf{M}$.

**Corollary 4.3.** *Assume that $f$ is M–smooth and $\mu$–strongly convex. Suppose that $\mathcal{D}$ is such that at each iteration standard unit basis vector $e_i$ is sampled with probability $p_i \propto \mathbf{M}_{ii}$. If we choose $\alpha = \frac{0.232}{\text{Trace}(\mathbf{M})}, \sigma = \frac{0.061}{\text{Trace}(\mathbf{M})}$, then $\mathbb{E}\left[\Psi^k\right] \leq \left(1 - \frac{0.117\mu}{\text{Trace}(\mathbf{M})}\right)^k\Psi^0$.*

The iteration complexities from Theorem 4.2 and Corollary 4.3 are summarized in Table 1. We also state that $\sigma, \alpha$ can be chosen so that (15) holds, and the rate from Theorem 4.2 coincides with the rate from Table 1. Theorem 4.2 and Corollary 4.3 hold even under a non-convex relaxation of strong convexity – Polyak-Łojasiewicz inequality: $\mu(f(x) - f(x^*)) \leq \frac{1}{2}\|\nabla f(x)\|_2^2$. Thus, SEGA works for a certain class of non-convex problems. For an overview on relaxations of strong convexity, see [23].

### 4.3 Accelerated method

In this section, we propose an accelerated (in the sense of Nesterov's method [31, 32]) version of `SEGA`, which we call `ASEGA`. The analogous accelerated `CD` method, in which a single coordinate is sampled in every iteration, was developed and analyzed in [3]. The general variant utilizing arbitrary sampling was developed and analyzed in [20].

---

**Algorithm 2:** `ASEGA`: Accelerated SEGA

---
1   **Initialize:** $x^0 = y^0 = z^0 \in \mathbb{R}^n$; $h^0 \in \mathbb{R}^n$; $S$; parameters $\alpha, \beta, \tau, \mu > 0$
2   **for** $k = 1, 2, \ldots$ **do**
3     $x^k = (1 - \tau)y^{k-1} + \tau z^{k-1}$
4     Sample $\mathbf{S}_k = \mathbf{I}_{S_k}$, where $S_k \sim S$, and compute $g^k, h^{k+1}$ according to (4), (6)
5     $y^k = x^k - \alpha \hat{\mathbf{P}}^{-1} g^k$
6     $z^k = \frac{1}{1+\beta\mu}(z^k + \beta\mu x^k - \beta g^k)$

---

The method and analysis is inspired by [2]. Due to space limitations and technicality of the content, we state the main theorem of this section in Appendix C.4. Here, we provide Corollary 4.4, which shows that Algorithm 2 with single coordinate sampling enjoys, up to a constant factor, the same convergence rate as state-of-the-art accelerated coordinate descent method `NUACDM` [3].

**Corollary 4.4.** *Let the sampling be defined as follows: $S = \{i\}$ w. p. $p_i \propto \sqrt{\mathbf{M}_{ii}}$, for $i \in [n]$. Then there exist acceleration parameters and a Lyapunov function $\Upsilon^k$ such that $f(y^k) - f(x^*) \leq \Upsilon^k$ and $\mathbb{E}\left[\Upsilon^k\right] \leq (1 - \tau)^k \Upsilon^0 = \left(1 - \mathcal{O}\left(\sqrt{\mu}/\sum_i \sqrt{\mathbf{M}_{ii}}\right)\right)^k \Upsilon^0$.*

The iteration complexity provided by Theorem C.5 and Corollary 4.4 are summarized in Table 1.

## 5 Experiments

In this section we perform numerical experiments to illustrate the potential of `SEGA`. Firstly, in Section 5.1, we compare it to projected gradient descent (PGD) algorithm. Then in Section 5.2, we study the performance of zeroth-order `SEGA` (when sketched gradients are being estimated through function value evaluations) and compare it to the analogous zeroth-order method. Lastly, in Section 5.3 we verify the claim from Remark 3.6 that in some applications, particular sketches and metric might lead to a significantly faster convergence. In the experiments where theory-supported stepsizes were used, we obtained them by precomputing strong convexity and smoothness measures.

### 5.1 Comparison to projected gradient

In this experiment, we show the potential superiority of our method to `PGD`. We consider the $\ell_2$ ball constrained problem ($R$ is the indicator function of the unit ball) with the oracle providing the sketched gradient in the random Gaussian direction. As we mentioned, a method moving in the gradient direction (analogue of `CD`), will not converge due as $R$ is not separable. Therefore, we can only compare against the projected gradient. In order to obtain the full gradient for `PGD`, one needs to gather $n$ sketched gradients and solve a corresponding linear system. As for $f$, we choose 4 different quadratics, see Table 2 (appendix). We stress that these are synthetic problems generated for the purpose of illustrating the potential of our method against a natural baseline. Figure 2 compares `SEGA` and `PGD` under various relative cost scenarios of solving the linear system compared to the cost of the oracle calls. The results show that `SEGA` significantly outperforms `PGD` as soon as solving the linear system is expensive, and is as fast as `PGD` even if solving the linear system comes for free.

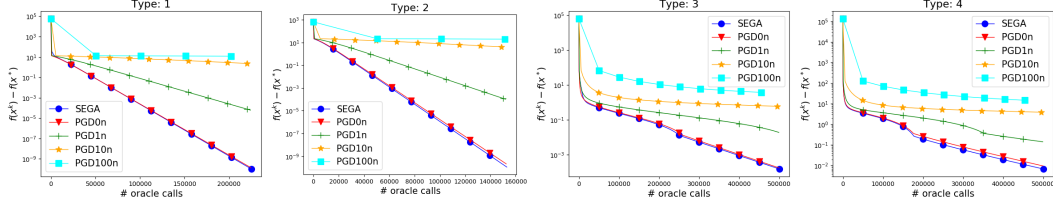

Figure 2: Convergence of `SEGA` and `PGD` on synthetic problems with $n = 500$. The indicator "Xn" in the label indicates the setting where the cost of solving linear system is $Xn$ times higher comparing to the oracle call. Recall that a linear system is solved after each $n$ oracle calls. Stepsizes $1/\lambda_{\max}(\mathbf{M})$ and $1/(n\lambda_{\max}(\mathbf{M}))$ were used for `PGD` and `SEGA`, respectively.

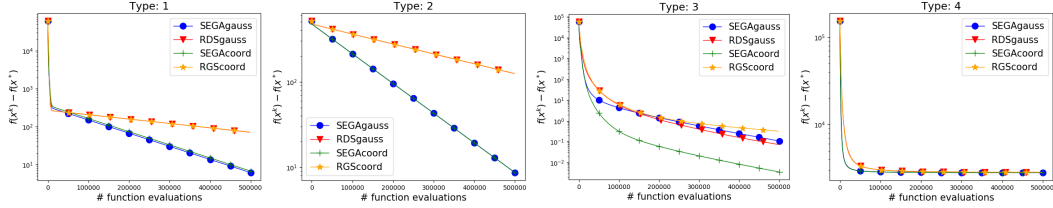

Figure 3: Comparison of `SEGA` and randomized direct search for various problems. Theory supported stepsizes were chosen for both methods. 500 dimensional problem.

## 5.2 Comparison to zeroth-order optimization methods

In this section, we compare `SEGA` to the *random direct search* (RDS) method [5] under a zeroth-order oracle and $R = 0$. For `SEGA`, we estimate the sketched gradient using finite differences. Note that RDS is a randomized version of the classical direct search method [21, 24, 25]. At iteration $k$, RDS moves to $\operatorname{argmin}\left(f(x^k + \alpha^k s^k), f(x^k - \alpha^k s^k), f(x^k)\right)$ for a random direction $s^k \sim \mathcal{D}$ and a suitable stepszie $\alpha^k$. For illustration, we choose $f$ to be a quadratic problem based on Table 2 and compare both Gaussian and coordinate sketches. Figure 3 shows that `SEGA` outperforms RDS.

## 5.3 Subspace `SEGA`: a more aggressive approach

As mentioned in Remark 3.6, well designed sketches are capable of exploiting structure of $f$ and lead to a better rate. We address this in detail in Appendix D where we develop and analyze a subspace variant of `SEGA`. To illustrate this phenomenon in a simple setting, we perform experiments for problem (1) with $f(x) = \|\mathbf{A}x - b\|^2$, where $b \in \mathbb{R}^d$ and $\mathbf{A} \in \mathbb{R}^{d \times n}$ has orthogonal rows, and with $R$ being the indicator function of the unit ball in $\mathbb{R}^n$. We assume that $n \gg d$. We compare two methods: `naiveSEGA`, which uses coordinate sketches, and `subspaceSEGA`, where sketches are chosen as rows of $\mathbf{A}$. Figure 4 indicates that `subspaceSEGA` outperforms `naiveSEGA` roughly by the factor $\frac{n}{d}$, as claimed in Appendix D.

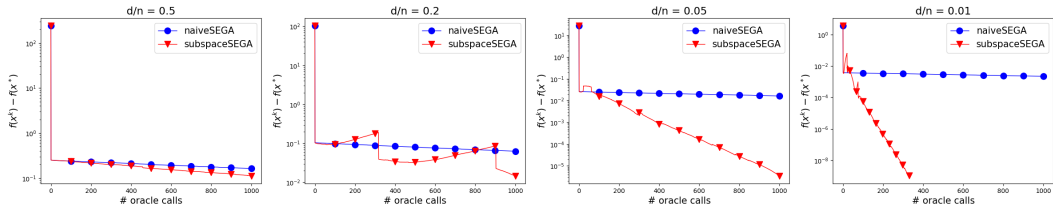

Figure 4: Comparison of `SEGA` with sketches from a correct subspace versus coordinate sketches `naiveSEGA`. Stepsize chosen according to theory. 1000 dimensional problem.

## Footnotes

[1]$f$ is $\mu$–strongly convex if $f(x) \geq f(y) + \langle \nabla f(y), x - y \rangle_{\mathbf{B}} + \frac{\mu}{2} \|x - y\|_{\mathbf{B}}^2$ for all $x, y \in \mathbb{R}^n$.

[2]Such a random variable may not exist. Some sufficient conditions are provided later.

[3]Forming vector $g$ and computing the prox.

[4]The standard $L$–smoothness assumption is a special case of $\mathbf{M}$–smoothness for $\mathbf{M} = L\mathbf{I}$, and hence is less general than both $\mathbf{M}$–smoothness and $\mathbf{Q}$–smoothness with respect to $\mathbf{B}$.

[5]There was recently introduced a notion of importance minibatch sampling for coordinate descent [20]. We state, without a proof, that SEGA allows for the same importance sampling as developed in the mentioned paper.

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
