[Supplementary Material]

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

[6]Strong convexity is not compatible with the assumption that $\mathbf{A}$ does not have full rank, so a different type of analysis using Polyak-Łojasiewicz inequality is required to give a formal justification. However, we proceed with the analysis anyway to build the intuition why this approach leads to better rates.

[7]Note that in the strong convexity inequality below the scalar product is without any additional metric unlike in other sections.

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

# Appendix

## A Conclusions and Extensions

### A.1 Conclusions

We proposed SEGA, a method for solving composite optimization problems under a novel stochastic linear first order oracle. SEGA is variance-reduced, and this is achieved via sketch-and-project updates of gradient estimates. We provided an analysis for smooth and strongly convex functions and general sketches, and a refined analysis for coordinate sketches. For coordinate sketches we also proposed an accelerated variant of SEGA, and our theory matches that of state-of-the-art CD methods. However, in contrast to CD, SEGA can be used for optimization problems with a *non-separable* proximal term. We develop a more aggressive subspace variant of the method—subspaceSEGA—which leads to improvements in the $n \gg d$ regime. In the Appendix we give several further results, including simplified and alternative analyses of SEGA in the coordinate setup from Example 2.1. Our experiments are encouraging and substantiate our theoretical predictions.

### A.2 Extensions

We now point to several potential extensions of our work.

**Speeding up the general method.** We believe that it should be possible to extend ASEGA to the general setup from Theorem 3.3. In such a case, it might be possible to design metric $\mathbf{B}$ and distribution of sketches $\mathcal{D}$ so as to outperform accelerated proximal gradient methods [33, 4].

**Biased gradient estimator.** Recall that SEGA uses unbiased gradient estimator $g^k$ for updating $x^k$ in a similar way JacSketch [19] or SAGA [10] do this for the stochastic finite sum optimization. Recently, a stochastic method for finite sum optimization using biased gradient estimators was proven to be more efficient [35]. Therefore, it might be possible to establish better properties for a biased variant of SEGA. To demonstrate the potential of this approach, in Appendix G.1 we plot the evolution of iterates for the very simple biased method which uses $h^k$ as an update for line 3 in Algorithm 1.

**Applications.** We believe that SEGA might work well in applications where a zeroth-order approach is inevitable, such as reinforcement learning. We therefore believe that SEGA might be an efficient proximal method in some reinforcement learning applications. We also believe that communication-efficient variants of SEGA can be used for distributed training of machine learning models. This is because SEGA can be adapted to communicate sparse model updates only.

## B Proofs for Section 3

**Lemma B.1.** *Suppose that* $\mathbf{B} = \mathbf{I}$ *and* $f$ *is twice differentiable. Assumption 3.1 is equivalent to Assumption 3.2 for* $\mathbf{Q} = \mathbf{M}^{-1}$.

**Proof:** We first establish that Assumption 3.1 implies Assumption 3.2. Summing up (10) for $(x, y)$ and $(y, x)$ yields

$$\langle \nabla f(x) - \nabla f(y), x - y \rangle \geq \|\nabla f(x) - \nabla f(y)\|_{\mathbf{Q}}^2.$$

Using Cauchy Schwartz inequality we obtain

$$\|x - y\|_{\mathbf{Q}^{-1}} \geq \|\nabla f(x) - \nabla f(y)\|_{\mathbf{Q}}.$$

By the mean value theorem, there is $z \in [x, y]$ such that $\nabla f(x) - \nabla f(y) = \nabla^2 f(z)(x - y)$. Thus

$$\|x - y\|_{\mathbf{Q}^{-1}} \geq \|x - y\|_{\nabla^2 f(z) \mathbf{Q} \nabla^2 f(z)}.$$

The above is equivalent to

$$\left(\nabla^2 f(z)\right)^{-\frac{1}{2}} \mathbf{Q}^{-1} \left(\nabla^2 f(z)\right)^{-\frac{1}{2}} \succeq \left(\nabla^2 f(z)\right)^{\frac{1}{2}} \mathbf{Q} \left(\nabla^2 f(z)\right)^{\frac{1}{2}}$$

Note that for any $\mathbf{M}' \succ 0$ we have $\mathbf{M}' \succeq \mathbf{M}^{-1}$ if and only if $\mathbf{M} \succeq \mathbf{I}$. Thus

$$\left(\nabla^2 f(z)\right)^{-\frac{1}{2}} \mathbf{Q}^{-1} \left(\nabla^2 f(z)\right)^{-\frac{1}{2}} \succeq \mathbf{I},$$

which is equivalent to $\mathbf{Q}^{-1} \succeq \nabla^2 f(z)$. To establish the other direction, denote $\phi(y) = f(y) - \langle \nabla f(x), y \rangle$. Clearly, $x$ is minimizer of $\phi$ and therefore we have

$$\phi(x) \leq \phi(x - \mathbf{M}^{-1} \nabla f(y)) \leq \phi(y) - \frac{1}{2} \|\nabla f(y)\|_{\mathbf{M}^{-1}}^2,$$

which is exactly (10) for $\mathbf{Q} = \mathbf{M}^{-1}$. $\qquad\square$

**Lemma B.2.** *For* $\mathbf{B} \succ 0$ *and* $\mathbf{Z}_k \overset{def}{=} \mathbf{S}_k (\mathbf{S}_k^\top \mathbf{B}^{-1} \mathbf{S}_k)^\dagger \mathbf{S}_k^\top$, *then*

$$\mathbf{Z}_k^\top \mathbf{B}^{-1} \mathbf{Z}_k = \mathbf{Z}_k. \tag{16}$$

**Proof:** It is a property of pseudo-inverse that for any matrices $\mathbf{A}, \mathbf{B}$ it holds $((\mathbf{A}\mathbf{B})^\dagger)^\top = (\mathbf{B}^\top \mathbf{A}^\top)^\dagger$, so $\mathbf{Z}_k^\top = \mathbf{Z}_k$. Moreover, we also know for any $\mathbf{A}$ that $\mathbf{A}^\dagger \mathbf{A} \mathbf{A}^\dagger = \mathbf{A}^\dagger$ and, thus,

$$\mathbf{Z}_k^\top \mathbf{B}^{-1} \mathbf{Z}_k = \mathbf{S}_k (\mathbf{S}_k^\top \mathbf{B}^{-1} \mathbf{S}_k)^\dagger \mathbf{S}_k^\top \mathbf{B}^{-1} \mathbf{S}_k (\mathbf{S}_k^\top \mathbf{B}^{-1} \mathbf{S}_k)^\dagger \mathbf{S}_k^\top = \mathbf{S}_k (\mathbf{S}_k^\top \mathbf{B}^{-1} \mathbf{S}_k)^\dagger \mathbf{S}_k^\top = \mathbf{Z}_k.$$

$\qquad\square$

## B.1  Proof of Theorem 3.3

We first state two lemmas which will be crucial for the analysis. They characterize key properties of the gradient learning process (4), (6) and will be used later to bound expected distances of both $h^{k+1}$ and $g^k$ from $\nabla f(x^*)$. The proofs are provided in Appendix B.2 and B.3 respectively

**Lemma B.3.** *For all* $v \in \mathbb{R}^n$ *we have*

$$\mathbb{E}_{\mathcal{D}} \left[ \|h^{k+1} - v\|_{\mathbf{B}}^2 \right] = \|h^k - v\|_{\mathbf{B} - \mathbb{E}_{\mathcal{D}}[\mathbf{Z}]}^2 + \|\nabla f(x^k) - v\|_{\mathbb{E}_{\mathcal{D}}[\mathbf{Z}]}^2. \tag{17}$$

**Lemma B.4.** *Let* $\mathbf{C} \overset{def}{=} \mathbb{E}_{\mathcal{D}} \left[ \theta^2 \mathbf{Z} \right]$. *Then for all* $v \in \mathbb{R}^n$ *we have*

$$\mathbb{E}_{\mathcal{D}} \left[ \|g^k - v\|_{\mathbf{B}}^2 \right] \leq 2 \|\nabla f(x^k) - v\|_{\mathbf{C}}^2 + 2 \|h^k - v\|_{\mathbf{C} - \mathbf{B}}^2.$$

For notational simplicity, it will be convenient to define Bregman divergence between $x$ and $y$:

$$D_f(x, y) \overset{def}{=} f(x) - f(y) - \langle \nabla f(y)), x - y \rangle_{\mathbf{B}}$$

We can now proceed with the proof of Theorem 3.3. Let us start with bounding the first term in the expression for $\Phi^{k+1}$. From Lemma B.4 and strong convexity it follows that

$$
\begin{aligned}
\mathbb{E}_{\mathcal{D}} \left[ \|x^{k+1} - x^*\|_{\mathbf{B}}^2 \right] &= \mathbb{E}_{\mathcal{D}} \left[ \| \operatorname{prox}_{\alpha R}(x^k - \alpha g^k) - \operatorname{prox}_{\alpha R}(x^* - \alpha \nabla f(x^*)) \|_{\mathbf{B}}^2 \right] \\
&\leq \mathbb{E}_{\mathcal{D}} \left[ \|x^k - \alpha g^k - (x^* - \alpha \nabla f(x^*))\|_{\mathbf{B}}^2 \right] \\
&= \|x^k - x^*\|_{\mathbf{B}}^2 - 2\alpha \mathbb{E}_{\mathcal{D}} \left[ (g^k - \nabla f(x^*))^\top \mathbf{B}(x^k - x^*) \right] \\
&\quad + \alpha^2 \mathbb{E}_{\mathcal{D}} \left[ \|g^k - \nabla f(x^*)\|_{\mathbf{B}}^2 \right] \\
&\leq \|x^k - x^*\|_{\mathbf{B}}^2 - 2\alpha (\nabla f(x^k) - \nabla f(x^*))^\top \mathbf{B}(x^k - x^*) \\
&\quad + 2\alpha^2 \|\nabla f(x^k) - \nabla f(x^*)\|_{\mathbf{C}}^2 + 2\alpha^2 \|h^k - \nabla f(x^*)\|_{\mathbf{C}-\mathbf{B}}^2 \\
&\leq \|x^k - x^*\|_{\mathbf{B}}^2 - \alpha \mu \|x^k - x^*\|_{\mathbf{B}}^2 - 2\alpha D_f(x^k, x^*) \\
&\quad + 2\alpha^2 \|\nabla f(x^k) - \nabla f(x^*)\|_{\mathbf{C}}^2 + 2\alpha^2 \|h^k - \nabla f(x^*)\|_{\mathbf{C}-\mathbf{B}}^2.
\end{aligned}
$$

Using Assumption 3.1 we get

$$-2\alpha D_f(x^k, x^*) \le -\alpha \|\nabla f(x^k) - \nabla f(x^*)\|_{\mathbf{Q}}^2.$$

As for the second term in $\Phi^{k+1}$, we have by Lemma B.3

$$\alpha\sigma\mathbb{E}_{\mathcal{D}}\left[\|h^{k+1} - \nabla f(x^*)\|_{\mathbf{B}}^2\right] = \alpha\sigma\|h^k - \nabla f(x^*)\|_{\mathbf{B}-\mathbb{E}_{\mathcal{D}}[\mathbf{Z}]}^2 + \alpha\sigma\|\nabla f(x^k) - \nabla f(x^*)\|_{\mathbb{E}_{\mathcal{D}}[\mathbf{Z}]}^2$$

Combining it into Lyapunov function $\Phi^k$,

$$
\begin{aligned}
\Phi^{k+1} \quad \le \quad & (1 - \alpha\mu)\|x^k - x^*\|_{\mathbf{B}}^2 + \alpha\sigma\|h^k - \nabla f(x^*)\|_{\mathbf{B}-\mathbb{E}_{\mathcal{D}}[\mathbf{Z}]}^2 + 2\alpha^2\|h^k - \nabla f(x^*)\|_{\mathbf{C}-\mathbf{B}}^2 \\
& + \alpha\sigma\|\nabla f(x^k) - \nabla f(x^*)\|_{\mathbb{E}_{\mathcal{D}}[\mathbf{Z}]}^2 + 2\alpha^2\|\nabla f(x^k) - \nabla f(x^*)\|_{\mathbf{C}}^2 - \alpha\|\nabla f(x^k) - \nabla f(x^*)\|_{\mathbf{Q}}^2.
\end{aligned}
$$

To see that this gives us the theorem's statement, consider first

$$\alpha\sigma\mathbb{E}_{\mathcal{D}}[\mathbf{Z}] + 2\alpha^2\mathbf{C} - \alpha\mathbf{Q} = 2\alpha(\alpha\mathbf{C} - \tfrac{1}{2}(\mathbf{Q} - \sigma\mathbb{E}_{\mathcal{D}}[\mathbf{Z}])) \le 0,$$

so we can drop norms related to $\nabla f(x^k) - \nabla f(x^*)$. Next, we have

$$
\begin{aligned}
\alpha\sigma(\mathbf{B} - \mathbb{E}_{\mathcal{D}}[\mathbf{Z}]) + 2\alpha^2(\mathbf{C} - \mathbf{B}) \quad &= \quad \alpha\left(\alpha(2(\mathbf{C} - \mathbf{B}) + \sigma\mu\mathbf{B}) - \mathbb{E}_{\mathcal{D}}[\mathbf{Z}]\right) + \sigma\alpha(1 - \alpha\mu)\mathbf{B} \\
&\le \quad \sigma\alpha(1 - \alpha\mu)\mathbf{B},
\end{aligned}
$$

which follows from our assumption on $\alpha$. $\qquad\square$

## B.2  Proof of Lemma B.3

**Proof:** Keeping in mind that $\mathbf{Z}_k^\top = \mathbf{Z}_k$ and $(\mathbf{B}^{-1})^\top = \mathbf{B}^{-1}$, we first write

$$
\begin{aligned}
\mathbb{E}_{\mathcal{D}}\left[\|h^{k+1} - v\|_{\mathbf{B}}^2\right] \quad &\overset{(8)}{=} \quad \mathbb{E}_{\mathcal{D}}\left[\|h^k + \mathbf{B}^{-1}\mathbf{Z}_k(\nabla f(x^k) - h^k) - v\|_{\mathbf{B}}^2\right] \\
&= \quad \mathbb{E}_{\mathcal{D}}\left[\|(\mathbf{I} - \mathbf{B}^{-1}\mathbf{Z}_k)(h^k - v) + \mathbf{B}^{-1}\mathbf{Z}_k(\nabla f(x^k) - v)\|_{\mathbf{B}}^2\right] \\
&= \quad \mathbb{E}_{\mathcal{D}}\left[\|(\mathbf{I} - \mathbf{B}^{-1}\mathbf{Z}_k)(h^k - v)\|_{\mathbf{B}}^2\right] + \mathbb{E}_{\mathcal{D}}\left[\|\mathbf{B}^{-1}\mathbf{Z}_k(\nabla f(x^k) - v)\|_{\mathbf{B}}^2\right] \\
&\qquad + 2(h^k - v)^\top \mathbb{E}_{\mathcal{D}}\left[(\mathbf{I} - \mathbf{B}^{-1}\mathbf{Z}_k)^\top \mathbf{B}\mathbf{B}^{-1}\mathbf{Z}_k\right](\nabla f(x^k) - v) \\
&= \quad (h^k - v)^\top \mathbb{E}_{\mathcal{D}}\left[(\mathbf{I} - \mathbf{B}^{-1}\mathbf{Z}_k)^\top \mathbf{B}(\mathbf{I} - \mathbf{B}^{-1}\mathbf{Z}_k)\right](h^k - v) \\
&\qquad + (\nabla f(x^k) - v)^\top \mathbb{E}_{\mathcal{D}}\left[\mathbf{Z}_k\mathbf{B}^{-1}\mathbf{B}\mathbf{B}^{-1}\mathbf{Z}_k\right](\nabla f(x^k) - v) \\
&\qquad + 2(h^k - v)^\top \mathbb{E}_{\mathcal{D}}\left[\mathbf{Z}_k - \mathbf{Z}_k\mathbf{B}^{-1}\mathbf{Z}_k\right](\nabla f(x^k) - v).
\end{aligned}
$$

By Lemma B.2 we have $\mathbf{Z}_k\mathbf{B}^{-1}\mathbf{Z}_k = \mathbf{Z}_k$, so the last term in the expression above is equal to 0. As for the other two, expanding the matrix factor in the first term leads to

$$
\begin{aligned}
\mathbb{E}_{\mathcal{D}}\left[(\mathbf{I} - \mathbf{B}^{-1}\mathbf{Z}_k)^\top \mathbf{B}(\mathbf{I} - \mathbf{B}^{-1}\mathbf{Z}_k)\right] \quad &= \quad \mathbb{E}_{\mathcal{D}}\left[(\mathbf{I} - \mathbf{Z}_k\mathbf{B}^{-1})\mathbf{B}(\mathbf{I} - \mathbf{B}^{-1}\mathbf{Z}_k)\right] \\
&= \quad \mathbb{E}_{\mathcal{D}}\left[\mathbf{B} - \mathbf{Z}_k\mathbf{B}^{-1}\mathbf{B} - \mathbf{B}\mathbf{B}^{-1}\mathbf{Z}_k + \mathbf{Z}_k\mathbf{B}^{-1}\mathbf{B}\mathbf{B}^{-1}\mathbf{Z}_k\right] \\
&= \quad \mathbf{B} - \mathbb{E}_{\mathcal{D}}[\mathbf{Z}_k].
\end{aligned}
$$

We, thereby, have derived

$$
\begin{aligned}
\mathbb{E}_{\mathcal{D}}\left[\|h^{k+1} - v\|_{\mathbf{B}}^2\right] \quad &= \quad (h^k - v)^\top (\mathbf{B} - \mathbb{E}_{\mathcal{D}}[\mathbf{Z}_k])(h^k - v) \\
&\qquad + (\nabla f(x^k) - v)^\top \mathbb{E}_{\mathcal{D}}\left[\mathbf{Z}_k\mathbf{B}^{-1}\mathbf{Z}_k\right](\nabla f(x^k) - v) \\
&= \quad \|h^k - v\|_{\mathbf{B}-\mathbb{E}_{\mathcal{D}}[\mathbf{Z}]}^2 + \|\nabla f(x^k) - v\|_{\mathbb{E}_{\mathcal{D}}[\mathbf{Z}]}^2.
\end{aligned}
$$

$\qquad\square$

## B.3 Proof of Lemma B.4

**Proof:** Throughout this proof, we will use without any mention that $\mathbf{Z}_k^\top = \mathbf{Z}_k$.

Writing $g^k - v = a + b$, where $a \overset{\text{def}}{=} (\mathbf{I} - \theta_k \mathbf{B}^{-1} \mathbf{Z}_k)(h^k - v)$ and $b \overset{\text{def}}{=} \theta_k \mathbf{B}^{-1} \mathbf{Z}_k (\nabla f(x^k) - v)$, we get $\|g^k\|_{\mathbf{B}}^2 \leq 2(\|a\|_{\mathbf{B}}^2 + \|b\|_{\mathbf{B}}^2)$. Using Lemma B.2 and the definition of $\theta_k$ yields

$$
\begin{aligned}
\mathbb{E}_{\mathcal{D}}\left[\|a\|_{\mathbf{B}}^2\right] &= \mathbb{E}_{\mathcal{D}}\left[\| \left(\mathbf{I} - \theta_k \mathbf{B}^{-1} \mathbf{Z}_k\right)(h^k - v)\|_{\mathbf{B}}^2\right] \\
&= (h^k - v)^\top \mathbb{E}_{\mathcal{D}}\left[\left(\mathbf{I} - \theta_k \mathbf{Z}_k \mathbf{B}^{-1}\right) \mathbf{B} \left(\mathbf{I} - \theta_k \mathbf{B}^{-1} \mathbf{Z}_k\right)\right](h^k - v) \\
&= (h^k - v)^\top \mathbb{E}_{\mathcal{D}}\left[\left(\mathbf{B} - \theta_k \mathbf{Z}_k \mathbf{B}^{-1} \mathbf{B} - \mathbf{B}\theta_k \mathbf{B}^{-1} \mathbf{Z}_k + \theta_k^2 \mathbf{Z}_k \mathbf{B}^{-1} \mathbf{B} \mathbf{B}^{-1} \mathbf{Z}_k\right)\right](h^k - v) \\
&= (h^k - v)^\top \mathbb{E}_{\mathcal{D}}\left[\left(\mathbf{B} - 2\mathbf{B} + \theta_k^2 \mathbf{Z}_k\right)\right](h^k - v) \\
&= \|h^k - v\|_{\mathbb{E}_{\mathcal{D}}[\theta^2 \mathbf{Z}] - \mathbf{B}}^2.
\end{aligned}
$$

Similarly, the second term in the upper bound on $g^k$ can be rewritten as

$$
\begin{aligned}
\mathbb{E}_{\mathcal{D}}\left[\|b\|_{\mathbf{B}}^2\right] &= \mathbb{E}_{\mathcal{D}}\left[\|\theta_k \mathbf{B}^{-1} \mathbf{Z}_k (\nabla f(x^k) - v)\|_{\mathbf{B}}^2\right] \\
&= (\nabla f(x^k) - v)^\top \mathbb{E}_{\mathcal{D}}\left[\theta_k^2 \mathbf{Z}_k \mathbf{B}^{-1} \mathbf{B} \mathbf{B}^{-1} \mathbf{Z}_k\right](\nabla f(x^k) - v) \\
&= \|\nabla f(x^k) - v\|_{\mathbf{C}}^2.
\end{aligned}
$$

Combining the pieces, we get the claim. $\qquad\square$

## C  Proofs for Section 4

### C.1  Technical Lemmas

We first start with an analogue of Lemma B.4 allowing for a norm different from $\|\cdot\|_{\mathbf{B}}$. We remark that matrix $\mathbf{Q}'$ in the lemma is not to be confused with the smoothness matrix $\mathbf{Q}$ from Assumption 3.1.

**Lemma C.1.** *Let $\mathbf{Q}' \succ 0$. The variance of $g^k$ as an estimator of $\nabla f(x^k)$ can be bounded as follows:*

$$
\frac{1}{2}\mathbb{E}_{\mathcal{D}}\left[\|g^k\|_{\mathbf{Q}'}^2\right] \leq \|h^k\|_{\hat{\mathbf{P}}^{-1}(\mathbf{P}\circ\mathbf{Q}')\hat{\mathbf{P}}^{-1} - \mathbf{Q}'}^2 + \|\nabla f(x^k)\|_{\hat{\mathbf{P}}^{-1}(\mathbf{P}\circ\mathbf{Q}')\hat{\mathbf{P}}^{-1}}^2. \tag{18}
$$

**Proof:** Denote $\mathbf{S}_k$ to be a matrix with columns $e_i$ for $i \in \text{Range}(\mathbf{S}_k)$. We first write

$$
g^k = \underbrace{h^k - \hat{\mathbf{P}}^{-1} \mathbf{S}_k \mathbf{S}_k^\top h^k}_{a} + \underbrace{\hat{\mathbf{P}}^{-1} \mathbf{S}_k \mathbf{S}_k^\top \nabla f(x^k)}_{b}.
$$

Let us bound the expectation of each term individually. The first term is equal to

$$
\begin{aligned}
\mathbb{E}_{\mathcal{D}}\left[\|a\|_{\mathbf{Q}'}^2\right] &= \mathbb{E}_{\mathcal{D}}\left[\left\|\left(\mathbf{I} - \hat{\mathbf{P}}^{-1} \mathbf{S}_k \mathbf{S}_k^\top\right) h^k\right\|_{\mathbf{Q}'}^2\right] \\
&= (h^k)^\top \mathbb{E}_{\mathcal{D}}\left[\left(\mathbf{I} - \hat{\mathbf{P}}^{-1} \mathbf{S}_k \mathbf{S}_k^\top\right)^\top \mathbf{Q}' \left(\mathbf{I} - \hat{\mathbf{P}}^{-1} \mathbf{S}_k \mathbf{S}_k^\top\right)\right] h^k \\
&= (h^k)^\top \mathbb{E}_{\mathcal{D}}\left[\left(\mathbf{Q}' - \hat{\mathbf{P}}^{-1} \mathbf{S}_k \mathbf{S}_k^\top \mathbf{Q}' - \mathbf{Q}' \mathbf{S}_k \mathbf{S}_k^\top \hat{\mathbf{P}}^{-1}\right)\right] h^k \\
&\qquad + (h^k)^\top \mathbb{E}_{\mathcal{D}}\left[\left(\hat{\mathbf{P}}^{-1} \mathbf{S}_k \mathbf{S}_k^\top \mathbf{Q}' \mathbf{S}_k \mathbf{S}_k^\top \hat{\mathbf{P}}^{-1}\right)\right] h^k \\
&= (h^k)^\top \left(\hat{\mathbf{P}}^{-1}(\mathbf{P} \circ \mathbf{Q}')\hat{\mathbf{P}}^{-1} - \mathbf{Q}'\right) h^k.
\end{aligned}
$$

The second term can be bounded as

$$
\begin{aligned}
\mathbb{E}_{\mathcal{D}}\left[\|b\|_{\mathbf{Q}'}^2\right] &= \mathbb{E}_{\mathcal{D}}\left[\left\|\hat{\mathbf{P}}^{-1} \mathbf{S}_k^\top \nabla f(x^k) \mathbf{S}_k\right\|_{\mathbf{Q}'}^2\right] = \mathbb{E}_{\mathcal{D}}\left[\|\nabla f(x^k)\|_{\hat{\mathbf{P}}^{-1} \mathbf{S}_k \mathbf{S}_k^\top \mathbf{Q}' \mathbf{S}_k \mathbf{S}_k^\top \hat{\mathbf{P}}^{-1}}^2\right] \\
&= \|\nabla f(x^k)\|_{\hat{\mathbf{P}}^{-1}(\mathbf{P}\circ\mathbf{Q}')\hat{\mathbf{P}}^{-1}}^2.
\end{aligned}
$$

It remains to combine the two bounds. $\qquad\square$

We also state the analogue of Lemma B.3, which allows for a different norm as well.

**Lemma C.2.** *For all diagonal* $\mathbf{D} \succ 0$ *we have*

$$\mathbb{E}_{\mathcal{D}}\left[\|h^{k+1}\|_{\mathbf{D}}^2\right] = \|h^k\|_{\mathbf{D}-\hat{\mathbf{P}}\mathbf{D}}^2 + \|\nabla f(x^k)\|_{\hat{\mathbf{P}}\mathbf{D}}^2. \tag{19}$$

**Proof:** Denote $\mathbf{S}_k$ to be a matrix with columns $e_i$ for $i \in \mathbf{S}_k$. We first write

$$h^{k+1} = h^k - \mathbf{S}_k\mathbf{S}_k^\top h^k + \mathbf{S}_k\mathbf{S}_k^\top \nabla f(x^k).$$

Therefore

$$
\begin{aligned}
\mathbb{E}_{\mathcal{D}}\left[\|h^{k+1}\|_{\mathbf{D}}^2\right] &= \mathbb{E}_{\mathcal{D}}\left[\left\|(\mathbf{I} - \mathbf{S}_k\mathbf{S}_k^\top)h^k + \mathbf{S}_k\mathbf{S}_k^\top \nabla f(x^k)\right\|_{\mathbf{D}}^2\right] \\
&= \mathbb{E}_{\mathcal{D}}\left[\left\|(\mathbf{I} - \mathbf{S}_k\mathbf{S}_k^\top)h^k\right\|_{\mathbf{D}}^2\right] + \mathbb{E}_{\mathcal{D}}\left[\left\|\mathbf{S}_k\mathbf{S}_k^\top \nabla f(x^k)\right\|_{\mathbf{D}}^2\right] \\
&\quad + 2\mathbb{E}_{\mathcal{D}}\left[{h^k}^\top(\mathbf{I} - \mathbf{S}_k\mathbf{S}_k^\top)\mathbf{D}\mathbf{S}_k\mathbf{S}_k^\top \nabla f(x^k)\right] \\
&= \|h^k\|_{\mathbf{D}-\hat{\mathbf{P}}\mathbf{D}}^2 + \|\nabla f(x^k)\|_{\hat{\mathbf{P}}\mathbf{D}}^2.
\end{aligned}
$$

$\square$

## C.2 Proof of Theorem 4.2

**Proof:** Throughout the proof, we will use the following Lyapunov function:

$$\Psi^k \overset{\text{def}}{=} f(x^k) - f(x^*) + \sigma\|h^k\|_{\mathbf{P}^{-1}}^2.$$

Following similar steps to what we did before, we obtain

$$
\begin{aligned}
\mathbb{E}\left[\Psi^{k+1}\right] &\overset{(11)}{\leq} f(x^k) - f(x^*) + \alpha\mathbb{E}\left[\langle\nabla f(x^k), g^k\rangle\right] + \frac{\alpha^2}{2}\mathbb{E}\left[\|g^k\|_{\mathbf{M}}^2\right] + \sigma\mathbb{E}\left[\|h^{k+1}\|_{\hat{\mathbf{P}}^{-1}}^2\right] \\
&= f(x^k) - f(x^*) - \alpha\|\nabla f(x^k)\|_2^2 + \frac{\alpha^2}{2}\mathbb{E}\left[\|g^k\|_{\mathbf{M}}^2\right] + \sigma\mathbb{E}\left[\|h^{k+1}\|_{\hat{\mathbf{P}}^{-1}}^2\right] \\
&\overset{(18)}{\leq} f(x^k) - f(x^*) - \alpha\|\nabla f(x^k)\|_2^2 + \alpha^2\|\nabla f(x^k)\|_{\hat{\mathbf{P}}^{-1}(\mathbf{P}\circ\mathbf{M})\hat{\mathbf{P}}^{-1}}^2 + \alpha^2\|h^k\|_{\hat{\mathbf{P}}^{-1}(\mathbf{P}\circ\mathbf{M})\hat{\mathbf{P}}^{-1}-\mathbf{M}}^2 \\
&\quad + \sigma\mathbb{E}\left[\|h^{k+1}\|_{\hat{\mathbf{P}}^{-1}}^2\right].
\end{aligned}
$$

This is the place where the ESO assumption comes into play. By applying it to the right-hand side of the bound above, we obtain

$$
\begin{aligned}
\mathbb{E}\left[\Psi^{k+1}\right] &\overset{(14)}{\leq} f(x^k) - f(x^*) - \alpha\|\nabla f(x^k)\|_2^2 + \alpha^2\|\nabla f(x^k)\|_{\hat{\mathbf{V}}\hat{\mathbf{P}}^{-1}}^2 + \alpha^2\|h^k\|_{\hat{\mathbf{V}}\hat{\mathbf{P}}^{-1}-\mathbf{M}}^2 \\
&\quad + \sigma\mathbb{E}\left[\|h^{k+1}\|_{\hat{\mathbf{P}}^{-1}}^2\right] \\
&\overset{(19)}{=} f(x^k) - f(x^*) - \alpha\|\nabla f(x^k)\|_2^2 + \alpha^2\|\nabla f(x^k)\|_{\hat{\mathbf{V}}\hat{\mathbf{P}}^{-1}}^2 + \alpha^2\|h^k\|_{\hat{\mathbf{V}}\hat{\mathbf{P}}^{-1}-\mathbf{M}}^2 \\
&\quad + \sigma\|\nabla f(x^k)\|_2^2 + \sigma\|h^k\|_{\hat{\mathbf{P}}^{-1}-\mathbf{I}}^2 \\
&= f(x^k) - f(x^*) - \left(\alpha - \alpha^2\max_i\frac{v_i}{p_i} - \sigma\right)\|\nabla f(x^k)\|_2^2 \\
&\quad + \|h^k\|_{\alpha^2(\hat{\mathbf{V}}\hat{\mathbf{P}}^{-1}-\mathbf{M})+\sigma(\hat{\mathbf{P}}^{-1}-\mathbf{I})}^2.
\end{aligned}
$$

Due to Polyak-Łojasiewicz inequality, we can further upper bound the last expression by

$$\left(1 - \left(\alpha - \alpha^2\max_i\frac{v_i}{p_i} - \sigma\right)\mu\right)(f(x^k) - f(x^*)) + \|h^k\|_{\alpha^2(\hat{\mathbf{V}}\mathbf{P}^{-1}-\mathbf{M})+\sigma(\mathbf{P}^{-1}-\mathbf{I})}^2.$$

To finish the proof, it remains to use (15). $\square$

## C.3 Proof of Corollary 4.3

The claim was obtained by choosing carefully $\alpha$ and $\sigma$ using numerical grid search. Note that by strong convexity we have $\mathbf{I} \succeq \mu \mathrm{Diag}(\mathbf{M})^{-1}$, so we can satisfy assumption (15). Then, the claim follows immediately noticing that we can also set $\hat{\mathbf{V}} = \mathrm{Diag}(\mathbf{M})$ while maintaining

$$\left( \alpha - \alpha^2 \max_i \frac{\mathbf{M}_{ii}}{p_i} - \sigma \right) \geq \frac{0.117}{\mathrm{Trace}(\mathbf{M})}.$$

## C.4 Accelerated `SEGA` with arbitrary sampling

Before establishing the main theorem, we first state two technical lemmas which will be crucial for the analysis. First one, Lemma C.3 provides a key inequality following from (6). The second one, Lemma C.4, analyzes update (5) and was technically established throughout the proof of Theorem 4.2. We include a proof of lemmas in Appendix C.5 and C.6 respectively.

**Lemma C.3.** *For every $u \in \mathbb{R}^n$ we have*

$$\beta\langle \nabla f(x^{k+1}), z^k - u \rangle - \frac{\beta\mu}{2}\|x^{k+1} - u\|_2^2$$
$$\leq \beta^2 \frac{1}{2}\mathbb{E}\left[\|g^k\|_2^2\right] + \frac{1}{2}\|z^k - u\|_2^2 - \frac{1+\beta\mu}{2}\mathbb{E}\left[\|z^{k+1} - u\|_2^2\right] \qquad (20)$$

**Lemma C.4.** *Letting $\eta(v, p) \stackrel{def}{=} \max_i \frac{\sqrt{v_i}}{p_i}$, we have*

$$f(x^{k+1}) - \mathbb{E}\left[f(y^{k+1})\right] + \|h^k\|_{\alpha^2(\hat{\mathbf{V}}\hat{\mathbf{P}}^{-3} - \hat{\mathbf{P}}^{-1}\mathbf{M}\hat{\mathbf{P}}^{-1})}^2 \geq \left(\alpha - \alpha^2\eta(v, p)^2\right)\|\nabla f(x^k)\|_{\hat{\mathbf{P}}^{-1}}^2. \qquad (21)$$

Now we state the main theorem of Section 4.3, providing a convergence rate of `ASEGA` (Algorithm 2) for arbitrary minibatch sampling. As we mentioned, the convergence rate is, up to a constant factor, same as state-of-the-art minibatch accelerated coordinate descent [20].

**Theorem C.5.** *Assume $\mathbf{M}$–smoothness and $\mu$–strong convexity and that $v$ satisfies (14). Denote*

$$\Upsilon^k \stackrel{def}{=} \frac{2}{75}\frac{\eta(v, p)^{-2}}{\tau^2}\left(\mathbb{E}\left[f(y^k)\right] - f(x^*)\right) + \frac{1+\beta\mu}{2}\mathbb{E}\left[\|z^k - x^*\|_2^2\right] + \sigma\mathbb{E}\left[\|h^k\|_{\hat{\mathbf{P}}^{-2}}^2\right]$$

*and choose*

$$c_1 = \max\left(1, \eta(v, p)^{-1}\frac{\sqrt{\mu}}{\min_i p_i}\right) \qquad (22)$$

$$\alpha = \frac{1}{5\eta(v, p)^2} \qquad (23)$$

$$\beta = \frac{2}{75\tau\eta(v, p)^2} \qquad (24)$$

$$\sigma = 5\beta^2 \qquad (25)$$

$$\tau = \frac{\sqrt{\frac{4}{9 \cdot 5^4}\eta(v, p)^{-4}\mu^2 + \frac{8}{75}\eta(v, p)^{-2}\mu} - \frac{2}{75}\eta(v, p)^{-2}\mu}{2} \qquad (26)$$

*Then, we have*

$$\mathbb{E}\left[\Upsilon^k\right] \leq \left(1 - c_1^{-1}\tau\right)^k \Upsilon^0.$$

**Proof:** The proof technique is inspired by [2]. First of all, let us see what strong convexity of $f$ gives us:

$$\beta\left(f(x^{k+1}) - f(x^*)\right) \leq \beta\langle \nabla f(x^{k+1}), x^{k+1} - x^* \rangle - \frac{\beta\mu}{2}\|x^* - x^{k+1}\|_2^2.$$

Thus, we are interested in finding an upper bound for the scalar product that appeared above. We have

$$\beta\langle\nabla f(x^{k+1}), z^k - u\rangle - \frac{\beta\mu}{2}\|x^{k+1} - u\|_2^2 + \sigma\mathbb{E}\left[\|h^{k+1}\|_{\hat{\mathbf{P}}^{-2}}^2\right]$$

$$\overset{(20)}{\leq} \beta^2\frac{1}{2}\mathbb{E}\left[\|g^k\|_2^2\right] + \frac{1}{2}\|z^k - u\|_2^2 - \frac{1+\beta\mu}{2}\mathbb{E}\left[\|z^{k+1} - u\|_2^2\right] + \sigma\mathbb{E}\left[\|h^{k+1}\|_{\hat{\mathbf{P}}^{-2}}^2\right].$$

Using the Lemmas introduced above, we can upper bound the norms of $g^k$ and $h^{k+1}$ by using norms of $h^k$ and $\nabla f(x^k)$ to get the following:

$$\beta^2\frac{1}{2}\mathbb{E}\left[\|g^k\|_2^2\right] + \sigma\mathbb{E}\left[\|h^{k+1}\|_{\hat{\mathbf{P}}^{-2}}^2\right]$$

$$\overset{(19)}{\leq} \beta^2\frac{1}{2}\mathbb{E}\left[\|g^k\|_2^2\right] + \sigma\|h^k\|_{\hat{\mathbf{P}}^{-2}-\hat{\mathbf{P}}^{-1}}^2 + \sigma\|\nabla f(x^k)\|_{\hat{\mathbf{P}}^{-1}}^2$$

$$\overset{(18)}{\leq} \beta^2\|h^k\|_{\hat{\mathbf{P}}^{-1}-\mathbf{I}}^2 + \beta^2\|\nabla f(x^k)\|_{\hat{\mathbf{P}}^{-1}}^2 + \sigma\|h^k\|_{\hat{\mathbf{P}}^{-2}-\hat{\mathbf{P}}^{-1}}^2 + \sigma\|\nabla f(x^k)\|_{\hat{\mathbf{P}}^{-1}}^2.$$

Now, let us get rid of $\nabla f(x^k)$ by using the gradients property from Lemma C.4:

$$\beta^2\frac{1}{2}\mathbb{E}\left[\|g^k\|_2^2\right] + \sigma\mathbb{E}\left[\|h^{k+1}\|_{\hat{\mathbf{P}}^{-2}}^2\right]$$

$$\overset{(21)}{\leq} \beta^2\|h^k\|_{\hat{\mathbf{P}}^{-1}-\mathbf{I}}^2 + \left(\beta^2 + \sigma\right)\frac{f(x^{k+1}) - f(y^{k+1}) + \|h^k\|_{\alpha^2(\hat{\mathbf{V}}\hat{\mathbf{P}}^{-3}-\hat{\mathbf{P}}^{-1}\mathbf{M}\hat{\mathbf{P}}^{-1})}^2}{\alpha - \alpha^2\eta(v,p)^2} + \sigma\|h^k\|_{\hat{\mathbf{P}}^{-2}-\hat{\mathbf{P}}^{-1}}^2$$

$$= \|h^k\|_{\beta^2(\hat{\mathbf{P}}^{-1}-\mathbf{I})+\frac{(\beta^2+\sigma)\alpha^2}{\alpha-\alpha^2\eta(v,p)^2}(\hat{\mathbf{V}}\hat{\mathbf{P}}^{-3}-\hat{\mathbf{P}}^{-1}\mathbf{M}\hat{\mathbf{P}}^{-1})+\sigma(\hat{\mathbf{P}}^{-2}-\hat{\mathbf{P}}^{-1})}^2$$

$$+ \frac{\beta^2 + \sigma}{\alpha - \alpha^2\eta(v,p)^2}(f(x^{k+1}) - \mathbb{E}\left[f(y^{k+1})\right])$$

$$\leq \|h^k\|_{\beta^2\hat{\mathbf{P}}^{-1}+\frac{(\beta^2+\sigma)\alpha^2}{\alpha-\alpha^2\eta(v,p)^2}\hat{\mathbf{V}}\hat{\mathbf{P}}^{-3}+\sigma(\hat{\mathbf{P}}^{-2}-\hat{\mathbf{P}}^{-1})}^2 + \frac{\beta^2 + \sigma}{\alpha - \alpha^2\eta(v,p)^2}(f(x^{k+1}) - \mathbb{E}\left[f(y^{k+1})\right]).$$

Plugging this into the bound with which we started the proof, we deduce

$$\beta\langle\nabla f(x^{k+1}), z^k - u\rangle - \frac{\beta\mu}{2}\|x^{k+1} - u\|_2^2 + \sigma\mathbb{E}\left[\|h^{k+1}\|_{\hat{\mathbf{P}}^{-2}}^2\right]$$

$$\leq \|h^k\|_{\beta^2\hat{\mathbf{P}}^{-1}+\frac{(\beta^2+\sigma)\alpha^2}{\alpha-\alpha^2\eta(v,p)^2}\hat{\mathbf{V}}\hat{\mathbf{P}}^{-3}+\sigma(\hat{\mathbf{P}}^{-2}-\hat{\mathbf{P}}^{-1})}^2$$

$$+ \frac{\beta^2 + \sigma}{\alpha - \alpha^2\eta(v,p)^2}(f(x^{k+1}) - \mathbb{E}\left[f(y^{k+1})\right]) + \frac{1}{2}\|z^k - u\|_2^2 - \frac{1+\beta\mu}{2}\mathbb{E}\left[\|z^{k+1} - u\|_2^2\right].$$

Recalling our first step, we get with a few rearrangements

$$\beta\left(f(x^{k+1}) - f(x^*)\right)$$

$$\leq \beta\langle\nabla f(x^{k+1}), x^{k+1} - x^*\rangle - \frac{\beta\mu}{2}\|x^* - x^{k+1}\|_2^2$$

$$= \beta\langle\nabla f(x^{k+1}), x^{k+1} - z^k\rangle + \beta\langle\nabla f(x^{k+1}), z^k - x^*\rangle - \frac{\beta\mu}{2}\|x^* - x^{k+1}\|_2^2$$

$$= \frac{(1-\tau)\beta}{\tau}\langle\nabla f(x^{k+1}), y^k - x^{k+1}\rangle + \beta\langle\nabla f(x^{k+1}), z^k - x^*\rangle - \frac{\beta\mu}{2}\|x^* - x^{k+1}\|_2^2$$

$$\leq \frac{(1-\tau)\beta}{\tau}\left(f(y^k) - f(x^{k+1})\right) + \|h^k\|_{\beta^2\hat{\mathbf{P}}^{-1}+\frac{(\beta^2+\sigma)\alpha^2}{\alpha-\alpha^2\eta(v,p)^2}\hat{\mathbf{V}}\hat{\mathbf{P}}^{-3}+\sigma(\hat{\mathbf{P}}^{-2}-\hat{\mathbf{P}}^{-1})}^2$$

$$+ \frac{\beta^2 + \sigma}{\alpha - \alpha^2\eta(v,p)^2}(f(x^{k+1}) - \mathbb{E}\left[f(y^{k+1})\right]) + \frac{1}{2}\|z^k - x^*\|_2^2$$

$$- \frac{1+\beta\mu}{2}\mathbb{E}\left[\|z^{k+1} - x^*\|_2^2\right] - \sigma\mathbb{E}\left[\|h^{k+1}\|_{\hat{\mathbf{P}}^{-2}}^2\right].$$

Let us choose $\sigma, \beta$ such that for some constant $c_2$ (which we choose at the end) we have

$$c_2\sigma = \beta^2, \qquad \beta = \frac{\alpha - \alpha^2\eta(v,p)^2}{(1+c_2^{-1})\tau}.$$

Consequently, we have

$$\frac{\alpha - \alpha^2 \eta(v,p)^2}{(1+c_2^{-1})\tau^2} \left(\mathbb{E}\left[f(y^{k+1})\right] - f(x^*)\right) + \frac{1+\beta\mu}{2}\mathbb{E}\left[\|z^{k+1} - x^*\|_2^2\right] + \sigma\mathbb{E}\left[\|h^{k+1}\|_{\hat{\mathbf{P}}^{-2}}^2\right]$$

$$\leq (1-\tau)\frac{\alpha - \alpha^2\eta(v,p)^2}{(1+c_2^{-1})\tau^2}\left(f(y^k) - f(x^*)\right) + \frac{1}{2}\|z^k - x^*\|_2^2$$

$$+\|h^k\|_{\left(\hat{\mathbf{P}}^{-1} - (1-c_2)\mathbf{I} + \frac{(1+c_2)\alpha^2}{\alpha - \alpha^2\eta(v,p)^2}\hat{\mathbf{V}}\hat{\mathbf{P}}^{-2}\right)\sigma\hat{\mathbf{P}}^{-1}}^2$$

Let us make a particular choice of $\alpha$, so that for some constant $c_3$ (which we choose at the end) we can obtain the equations below:

$$\alpha = \frac{1}{c_3\eta(v,p)^2} \quad \Rightarrow \quad \alpha - \alpha^2\eta(v,p)^2 = \frac{c_3 - 1}{c_3^2}\eta(v,p)^{-2}, \quad \frac{\alpha^2}{\alpha - \alpha^2\eta(v,p)^2} = \frac{1}{(c_3 - 1)\eta(v,p)^2}.$$

Thus

$$\frac{\frac{c_3-1}{c_3^2}\eta(v,p)^{-2}}{(1+c_2^{-1})\tau^2}\left(\mathbb{E}\left[f(y^{k+1})\right] - f(x^*)\right) + \frac{1+\beta\mu}{2}\mathbb{E}\left[\|z^{k+1} - x^*\|_2^2\right] + \sigma\mathbb{E}\left[\|h^{k+1}\|_{\hat{\mathbf{P}}^{-2}}^2\right]$$

$$\leq (1-\tau)\frac{\frac{c_3-1}{c_3^2}\eta(v,p)^{-2}}{(1+c_2^{-1})\tau^2}\left(f(y^k) - f(x^*)\right) + \frac{1}{2}\|z^k - x^*\|_2^2$$

$$+\|h^k\|_{\left(\hat{\mathbf{P}}^{-1} - (1-c_2)\mathbf{I} + \frac{(1+c_2)}{(c_3-1)\eta(v,p)^2}\hat{\mathbf{V}}\hat{\mathbf{P}}^{-2}\right)\sigma\hat{\mathbf{P}}^{-1}}^2.$$

Using the definition of $\eta(v,p)$, one can see that the above gives

$$\frac{\frac{c_3-1}{c_3^2}\eta(v,p)^{-2}}{(1+c_2^{-1})\tau^2}\left(\mathbb{E}\left[f(y^{k+1})\right] - f(x^*)\right) + \frac{1+\beta\mu}{2}\mathbb{E}\left[\|z^{k+1} - x^*\|_2^2\right] + \sigma\mathbb{E}\left[\|h^{k+1}\|_{\hat{\mathbf{P}}^{-2}}^2\right]$$

$$\leq (1-\tau)\frac{\frac{c_3-1}{c_3^2}\eta(v,p)^{-2}}{(1+c_2^{-1})\tau^2}\left(f(y^k) - f(x^*)\right) + \frac{1}{2}\|z^k - x^*\|_2^2 + \|h^k\|_{\left(\hat{\mathbf{P}}^{-1} - (1-c_2)\mathbf{I} + \frac{1+c_2}{c_3-1}\mathbf{I}\right)\sigma\hat{\mathbf{P}}^{-1}}^2.$$

To get the convergence rate, we shall establish

$$\left(1 - c_2 - \frac{1+c_2}{c_3-1}\right)c_1\mathbf{I} \succeq \tau\hat{\mathbf{P}}^{-1} \tag{27}$$

and

$$1 + \beta\mu \geq \frac{1}{1-\tau}. \tag{28}$$

To this end, let us recall that

$$\beta = \frac{c_3 - 1}{c_2^2}\eta(v,p)^{-2}\tau^{-1}\frac{1}{1+c_2^{-1}}.$$

Now we would like to set equality in (28), which yields

$$0 = \tau^2 + \frac{c_3 - 1}{c_2^2}\eta(v,p)^{-2}\frac{1}{1+c_2^{-1}}\mu\tau - \frac{c_3-1}{c_2^2}\eta(v,p)^{-2}\frac{1}{1+c_2^{-1}}\mu = 0.$$

This, in turn, implies

$$\tau = \frac{\sqrt{\left(\frac{c_3-1}{c_2^2}\right)^2\eta(v,p)^{-4}\frac{1}{(1+c_2^{-1})^2}\mu^2 + 4\frac{c_3-1}{c_2^2}\eta(v,p)^{-2}\frac{1}{1+c_2^{-1}}\mu} - \frac{c_3-1}{c_2^2}\eta(v,p)^{-2}\frac{1}{1+c_2^{-1}}\mu}{2}$$

$$= \mathcal{O}\left(\sqrt{\frac{c_3-1}{c_2^2}}\frac{1}{\sqrt{1+c_2^{-1}}}\eta(v,p)^{-1}\sqrt{\mu}\right).$$

Notice that for any $c \leq 1$ we have $\frac{\sqrt{c^2+4c}-c}{2} \leq \sqrt{c}$ and therefore

$$\tau \leq \sqrt{\frac{c_3-1}{c_2^2}} \eta(v,p)^{-1} \frac{1}{\sqrt{1+c_2^{-1}}} \sqrt{\mu}. \tag{29}$$

Using this inequality and a particular choice of constants, we can upper bound $\mathbf{P}^{-1}$ by a matrix proportional to identity as shown below:

$$\tau \hat{\mathbf{P}}^{-1} \overset{(29)}{\preceq} \sqrt{\frac{c_3-1}{c_2^2}} \eta(v,p)^{-1} \frac{1}{\sqrt{1+c_2^{-1}}} \sqrt{\mu} \hat{\mathbf{P}}^{-1}$$

$$\preceq \sqrt{\frac{c_3-1}{c_2^2}} \eta(v,p)^{-1} \frac{1}{\sqrt{1+c_2^{-1}}} \frac{\sqrt{\mu}}{\min_i p_i} \mathbf{I}$$

$$\overset{(22)}{\preceq} \sqrt{\frac{c_3-1}{c_2^2}} \frac{1}{\sqrt{1+c_2^{-1}}} c_1 \mathbf{I}$$

$$\overset{(*)}{\preceq} \left(1 - c_2 - \frac{1+c_2}{c_3-1}\right) c_1 \mathbf{I},$$

which is exactly (27). Above, $(*)$ holds for choice $c_3 = 5$ and $c_2 = \frac{1}{5}$. It remains to verify that (23), (24), (25) and (26) indeed correspond to our derivations. $\qquad \square$

We also mention, without a proof, that acceleration parameters can be chosen in general such that $c_1$ can be lower bounded by constant and therefore the rate from Theorem C.5 coincides with the rate from Table 1. Corollary 4.4 is in fact a weaker result of that type.

### C.4.1 Proof of Corollary 4.4

It suffices to verify that one can choose $v = \mathrm{Diag}(\mathbf{M})$ in (14) and that due to $p_i \propto \sqrt{\mathbf{M}_{ii}}$ we have $c_1 = 1$.

### C.5 Proof of Lemma C.3

**Proof:** Firstly (6), is equivalent to

$$z^{k+1} = \underset{z}{\arg\min} \, \psi^k(z) \overset{\mathrm{def}}{=} \frac{1}{2}\|z - z^k\|_2^2 + \beta\langle g^k, z\rangle + \frac{\beta\mu}{2}\|z - x^{k+1}\|_2^2.$$

Therefore, we have for every $u$

$$0 = \langle \nabla\psi^k(z^{k+1}), z^{k+1} - u\rangle$$
$$= \langle z^{k+1} - z^k, z^{k+1} - u\rangle + \beta\langle g^k, z^{k+1} - u\rangle + \beta\mu\langle z^{k+1} - x^{k+1}, z^{k+1} - u\rangle. \tag{30}$$

Next, by generalized Pythagorean theorem we have

$$\langle z^{k+1} - z^k, z^{k+1} - u\rangle = \frac{1}{2}\|z^k - z^{k+1}\|_2^2 - \frac{1}{2}\|z^k - u\|_2^2 + \frac{1}{2}\|u - z^{k+1}\|_2^2 \tag{31}$$

and

$$\langle z^{k+1} - x^{k+1}, z^{k+1} - u\rangle = \frac{1}{2}\|x^{k+1} - z^{k+1}\|_2^2 - \frac{1}{2}\|x^{k+1} - u\|_2^2 + \frac{1}{2}\|u - z^{k+1}\|_2^2. \tag{32}$$

Plugging (31) and (32) into (30) we obtain

$$\beta\langle g^k, z^k - u\rangle - \frac{\beta\mu}{2}\|x^{k+1} - u\|_2^2$$

$$\leq \beta\langle g^k, z^k - z^{k+1}\rangle - \frac{1}{2}\|z^k - z^{k+1}\|_2^2 + \frac{1}{2}\|z^k - u\|_2^2 - \frac{1+\beta\mu}{2}\|z^{k+1} - u\|_2^2$$

$$\overset{(*)}{\leq} \frac{\beta^2}{2}\|g^k\|_2^2 + \frac{1}{2}\|z^k - u\|_2^2 - \frac{1+\beta\mu}{2}\|z^{k+1} - u\|_2^2.$$

The step marked by $(*)$ holds due to Cauchy-Schwartz inequality. It remains to take the expectation conditioned on $x^{k+1}$ and use (7). $\qquad\square$

## C.6 Proof of Lemma C.4

**Proof:** The shortest, although not the most intuitive, way to write the proof is to put matrix factor into norms. Apart from this trick, the proof is quite simple consists of applying smoothness followed by ESO:

$$
\begin{aligned}
\mathbb{E}\left[f(y^{k+1})\right] - f(x^{k+1}) &\overset{(11)}{\leq} -\alpha\mathbb{E}\left[\langle\nabla f(x^k),\hat{\mathbf{P}}^{-1}g^k\rangle\right] + \frac{\alpha^2}{2}\mathbb{E}\left[\|\hat{\mathbf{P}}^{-1}g^k\|_{\mathbf{M}}^2\right] \\
&= -\alpha\|\nabla f(x^k)\|_{\hat{\mathbf{P}}^{-1}}^2 + \frac{\alpha^2}{2}\mathbb{E}\left[\|g^k\|_{\hat{\mathbf{P}}^{-1}\mathbf{M}\hat{\mathbf{P}}^{-1}}\right] \\
&\overset{(18)}{\leq} -\alpha\|\nabla f(x^k)\|_{\hat{\mathbf{P}}^{-1}}^2 + \alpha^2\|\nabla f(x^k)\|_{\hat{\mathbf{P}}^{-1}(\mathbf{P}\circ\hat{\mathbf{P}}^{-1}\mathbf{M}\hat{\mathbf{P}}^{-1})\hat{\mathbf{P}}^{-1}}^2 \\
&\quad +\alpha^2\|h^k\|_{\hat{\mathbf{P}}^{-1}(\mathbf{P}\circ\hat{\mathbf{P}}^{-1}\mathbf{M}\hat{\mathbf{P}}^{-1})\hat{\mathbf{P}}^{-1}-\hat{\mathbf{P}}^{-1}\mathbf{M}\hat{\mathbf{P}}^{-1}}^2 \\
&= -\alpha\|\nabla f(x^k)\|_{\hat{\mathbf{P}}^{-1}}^2 + \alpha^2\|\nabla f(x^k)\|_{\hat{\mathbf{P}}^{-2}(\mathbf{P}\circ\mathbf{M})\hat{\mathbf{P}}^{-2}}^2 \\
&\quad +\alpha^2\|h^k\|_{\hat{\mathbf{P}}^{-2}(\mathbf{P}\circ\mathbf{M})\hat{\mathbf{P}}^{-2}-\hat{\mathbf{P}}^{-1}\mathbf{M}\hat{\mathbf{P}}^{-1}}^2 \\
&\overset{(14)}{\leq} -\alpha\|\nabla f(x^k)\|_{\hat{\mathbf{P}}^{-1}}^2 + \alpha^2\|\nabla f(x^k)\|_{\mathbf{V}\hat{\mathbf{P}}^{-3}}^2 \\
&\quad +\alpha^2\|h^k\|_{\mathbf{V}\hat{\mathbf{P}}^{-3}-\hat{\mathbf{P}}^{-1}\mathbf{M}\hat{\mathbf{P}}^{-1}}^2 \\
&\leq -\left(\alpha - \alpha^2\max_i\frac{v_i}{p_i^2}\right)\|f(x^k)\|_{\hat{\mathbf{P}}^{-1}}^2 + \alpha^2\|h^k\|_{\mathbf{V}\hat{\mathbf{P}}^{-3}-\hat{\mathbf{P}}^{-1}\mathbf{M}\hat{\mathbf{P}}^{-1}}^2.
\end{aligned}
$$

$\qquad\square$

# D Subspace `SEGA`: a More Aggressive Approach

In this section we describe a *more aggressive* variant of `SEGA`, one that exploits the fact that the gradients of $f$ lie in a lower dimensional subspace if this is indeed the case.

In particular, assume that $F(x) = f(x) + R(x)$ and

$$f(x) = \phi(\mathbf{A}x),$$

where $\mathbf{A} \in \mathbb{R}^{m\times n}$[6]. Note that $\nabla f(x)$ lies in Range $\left(\mathbf{A}^\top\right)$. There are situations where the dimension of Range $\left(\mathbf{A}^\top\right)$ is much smaller than $n$. For instance, this happens when $m \ll n$. However, standard coordinate descent methods still move around in directions $e_i \in \mathbb{R}^n$ for all $i$. We can modify the gradient sketch method to force our gradient estimate to lie in Range $\left(\mathbf{A}^\top\right)$, hoping that this will lead to faster convergence.

## D.1 The algorithm

Let $x^k$ be the current iterate, and let $h^k$ be the current estimate of the gradient of $f$. Assume that the sketch $\mathbf{S}_k^\top\nabla f(x^k)$ is available. We can now define $h^{k+1}$ through the following modified sketch-and-project process:

$$
\begin{aligned}
h^{k+1} &= \arg\min_{h\in\mathbb{R}^n} \|h - h^k\|_{\mathbf{B}}^2 \\
&\text{subject to} \quad \mathbf{S}_k^\top h = \mathbf{S}_k^\top\nabla f(x^k), \\
&\qquad\qquad\quad h \in \text{Range}\left(\mathbf{A}^\top\right).
\end{aligned}
\tag{33}
$$

Before proceeding further, we note that there are such sketches and metric (as discussed in Section D.4) which keep $h \in \text{Range}\left(\mathbf{A}^\top\right)$ implicitly, and therefore one might omit the extra constraint in such case. In fact, the mentioned sketches also lead to a faster convergence, which is the main takeaway from this section.

Standard arguments reveal that the closed-form solution of (33) is

$$h^{k+1} = \mathbf{H} \left(h^k - \mathbf{B}^{-1}\mathbf{S}_k(\mathbf{S}_k^\top \mathbf{H}\mathbf{B}^{-1}\mathbf{S}_k)^\dagger \mathbf{S}_k^\top(\mathbf{H}h^k - \nabla f(x^k))\right), \tag{34}$$

where

$$\mathbf{H} \overset{\text{def}}{=} \mathbf{A}^\top(\mathbf{A}\mathbf{B}\mathbf{A}^\top)^\dagger \mathbf{A}\mathbf{B} \tag{35}$$

is the projector onto $\text{Range}\left(\mathbf{A}^\top\right)$. A quick sanity check reveals that this gives the same formula as (4) in the case where $\text{Range}\left(\mathbf{A}^\top\right) = \mathbb{R}^n$. We can also write

$$h^{k+1} = \mathbf{H}h^k - \mathbf{H}\mathbf{B}^{-1}\mathbf{Z}_k(\mathbf{H}h^k - \nabla f(x^k)) = \left(\mathbf{I} - \mathbf{H}\mathbf{B}^{-1}\mathbf{Z}_k\right)\mathbf{H}h^k + \mathbf{H}\mathbf{B}^{-1}\mathbf{Z}_k\nabla f(x^k), \tag{36}$$

where

$$\mathbf{Z}_k \overset{\text{def}}{=} \mathbf{S}_k(\mathbf{S}_k^\top \mathbf{H}\mathbf{B}^{-1}\mathbf{S}_k)^\dagger \mathbf{S}_k^\top. \tag{37}$$

Assume that $\theta_k$ is chosen in such a way that

$$\mathbb{E}_{\mathcal{D}}\left[\theta_k \mathbf{Z}_k\right] = \mathbf{B}.$$

Then, the following estimate of $\nabla f(x^k)$

$$g^k \overset{\text{def}}{=} \mathbf{H}h^k + \theta_k \mathbf{H}\mathbf{B}^{-1}\mathbf{Z}_k(\nabla f(x^k) - \mathbf{H}h^k) \tag{38}$$

is unbiased, i.e. $\mathbb{E}_{\mathcal{D}}\left[g^k\right] = \nabla f(x^k)$. After evaluating $g^k$, we perform the same step as in SEGA:

$$x^{k+1} = \text{prox}_{\alpha R}(x^k - \alpha g^k).$$

By inspecting (33), (35) and (38), we get the following simple observation.

**Lemma D.1.** *If $h^0 \in \text{Range}\left(\mathbf{A}^\top\right)$, then $h^k, g^k \in \text{Range}\left(\mathbf{A}^\top\right)$ for all $k$.*

Consequently, if $h^0 \in \text{Range}\left(\mathbf{A}^\top\right)$, (34) simplifies to

$$h^{k+1} = h^k - \mathbf{H}\mathbf{B}^{-1}\mathbf{S}_k(\mathbf{S}_k^\top \mathbf{H}\mathbf{B}^{-1}\mathbf{S}_k)^\dagger \mathbf{S}_k^\top(h^k - \nabla f(x^k)) \tag{39}$$

and (38) simplifies to

$$g^k \overset{\text{def}}{=} h^k + \theta_k \mathbf{H}\mathbf{B}^{-1}\mathbf{Z}_k(\nabla f(x^k) - h^k). \tag{40}$$

**Example D.2** (Coordinate sketch). *Consider $\mathbf{B} = \mathbf{I}$ and the choice of $\mathcal{D}$ given by $\mathbf{S} = e_i$ with probability $p_i > 0$. Then we can choose the bias-correcting random variable as $\theta = \theta(s) = \frac{w_i}{p_i}$, where $w_i \overset{\text{def}}{=} \|\mathbf{H}e_i\|_2^2 = e_i^\top \mathbf{H}e_i$. Indeed, with this choice, (5) is satisfied. For simplicity, further choose $p_i = 1/n$ for all $i$. We then have*

$$h^{k+1} = h^k - \frac{e_i^\top h^k - e_i^\top \nabla f(x^k)}{w_i}\mathbf{H}e_i = \left(\mathbf{I} - \frac{\mathbf{H}e_i e_i^\top}{w_i}\right)h^k + \frac{\mathbf{H}e_i e_i^\top}{w_i}\nabla f(x^k) \tag{41}$$

*and* (40) *simplifies to*

$$g^k \overset{\text{def}}{=} (1 - \theta_k)h^k + \theta_k h^{k+1} = h^k + n\mathbf{H}e_i e_i^\top \left(\nabla f(x^k) - h^k\right). \tag{42}$$

## D.2 Lemmas

All theory provided in this subsection is, in fact, a straightforward generalization of our non-subspace results. The reader can recognize similarities in both statements and proofs with that of previous sections.

**Lemma D.3.** *Define* $\mathbf{Z}_k$ *and* $\mathbf{H}$ *as in equations* (37) *and* (35). *Then* $\mathbf{Z}_k$ *is symmetric,* $\mathbf{Z}_k \mathbf{H} \mathbf{B}^{-1} \mathbf{Z}_k = \mathbf{Z}_k$, $\mathbf{H}^2 = \mathbf{H}$ *and* $\mathbf{H} \mathbf{B}^{-1} = \mathbf{B}^{-1} \mathbf{H}^\top$.

**Proof:** The symmetry of $\mathbf{Z}_k$ follows from its definition. The second statement is a corollary of the equations $((\mathbf{A}_1 \mathbf{A}_2)^\dagger)^\top = (\mathbf{A}_2^\top \mathbf{A}_1^\top)^\dagger$ and $\mathbf{A}_1^\dagger \mathbf{A}_1 \mathbf{A}_1^\dagger = \mathbf{A}_1^\dagger$, which are true for any matrices $\mathbf{A}_1, \mathbf{A}_2$. Finally, the last two rules follow directly from the definition of $\mathbf{H}$ and the property $\mathbf{A}_1^\dagger \mathbf{A}_1 \mathbf{A}_1^\dagger = \mathbf{A}_1^\dagger$. $\square$

**Lemma D.4.** *Assume* $h^k \in Range\left(\mathbf{A}^\top\right)$. *Then*
$$\mathbb{E}_{\mathcal{D}}\left[\|h^{k+1} - v\|_{\mathbf{B}}^2\right] = \|h^k - v\|_{\mathbf{B} - \mathbb{E}_{\mathcal{D}}[\mathbf{Z}]}^2 + \|\nabla f(x^k) - v\|_{\mathbb{E}_{\mathcal{D}}[\mathbf{Z}]}^2$$
*for any vector* $v \in Range\left(\mathbf{A}^\top\right)$.

**Proof:** By Lemma D.3 we can rewrite $\mathbf{H} \mathbf{B}^{-1}$ as $\mathbf{B}^{-1} \mathbf{H}^\top$, so
$$
\begin{aligned}
\mathbb{E}_{\mathcal{D}}\left[\|h^{k+1} - v\|_{\mathbf{B}}^2\right] &\overset{(36)}{=} \mathbb{E}_{\mathcal{D}}\left[\left\|h^k - \mathbf{H}\mathbf{B}^{-1}\mathbf{Z}_k(h^k - \nabla f(x^k)) - v\right\|_{\mathbf{B}}^2\right] \\
&= \mathbb{E}_{\mathcal{D}}\left[\left\|\left(\mathbf{I} - \mathbf{H}\mathbf{B}^{-1}\mathbf{Z}_k\right)(h^k - v) + \mathbf{H}\mathbf{B}^{-1}\mathbf{Z}_k(\nabla f(x^k) - v)\right\|_{\mathbf{B}}^2\right] \\
&= \mathbb{E}_{\mathcal{D}}\left[\left\|\left(\mathbf{I} - \mathbf{B}^{-1}\mathbf{H}^\top\mathbf{Z}_k\right)(h^k - v) + \mathbf{H}\mathbf{B}^{-1}\mathbf{Z}_k(\nabla f(x^k) - v)\right\|_{\mathbf{B}}^2\right] \\
&= \mathbb{E}_{\mathcal{D}}\left[\left\|\left(\mathbf{I} - \mathbf{B}^{-1}\mathbf{H}^\top\mathbf{Z}_k\right)(h^k - v)\right\|_{\mathbf{B}}^2\right] + \mathbb{E}_{\mathcal{D}}\left[\left\|\mathbf{H}\mathbf{B}^{-1}\mathbf{Z}_k(\nabla f(x^k) - v)\right\|_{\mathbf{B}}^2\right] \\
&\quad + 2(h^k - v)^\top \mathbb{E}_{\mathcal{D}}\left[\left(\mathbf{I} - \mathbf{B}^{-1}\mathbf{H}^\top\mathbf{Z}_k\right)^\top \mathbf{B}\mathbf{H}\mathbf{B}^{-1}\mathbf{Z}_k\right](\nabla f(x^k) - v) \\
&= (h^k - v)^\top \mathbb{E}_{\mathcal{D}}\left[\left(\mathbf{I} - \mathbf{B}^{-1}\mathbf{H}^\top\mathbf{Z}_k\right)^\top \mathbf{B}\left(\mathbf{I} - \mathbf{H}\mathbf{B}^{-1}\mathbf{Z}_k\right)\right](h^k - v) \\
&\quad + (\nabla f(x^k) - v)^\top \mathbb{E}_{\mathcal{D}}\left[\mathbf{Z}_k\mathbf{B}^{-1}\mathbf{H}^\top\mathbf{B}\mathbf{H}\mathbf{B}^{-1}\mathbf{Z}_k\right](\nabla f(x^k) - v) \\
&\quad + 2(h^k - v)^\top \mathbb{E}_{\mathcal{D}}\left[\mathbf{B}\mathbf{H}\mathbf{B}^{-1}\mathbf{Z}_k - \mathbf{Z}_k\mathbf{H}\mathbf{H}\mathbf{B}^{-1}\mathbf{Z}_k\right](\nabla f(x^k) - v). \quad (43)
\end{aligned}
$$
By Lemma D.3 we have
$$\mathbf{Z}_k\mathbf{H}\mathbf{H}\mathbf{B}^{-1}\mathbf{Z}_k = \mathbf{Z}_k\mathbf{H}\mathbf{B}^{-1}\mathbf{Z}_k = \mathbf{Z}_k,$$
so the last term in (43) is equal to 0. As for the other two, expanding the matrix factor in the first term leads to
$$
\begin{aligned}
\left(\mathbf{I} - \mathbf{B}^{-1}\mathbf{H}^\top\mathbf{Z}_k\right)^\top \mathbf{B}\left(\mathbf{I} - \mathbf{H}\mathbf{B}^{-1}\mathbf{Z}_k\right) &= \left(\mathbf{I} - \mathbf{Z}_k\mathbf{H}\mathbf{B}^{-1}\right)\mathbf{B}\left(\mathbf{I} - \mathbf{H}\mathbf{B}^{-1}\mathbf{Z}_k\right) \\
&= \mathbf{B} - \mathbf{Z}_k\mathbf{H}\mathbf{B}^{-1}\mathbf{B} - \mathbf{B}\mathbf{B}^{-1}\mathbf{H}^\top\mathbf{Z}_k + \mathbf{Z}_k\mathbf{H}\mathbf{B}^{-1}\mathbf{B}\mathbf{H}\mathbf{B}^{-1}\mathbf{Z}_k \\
&= \mathbf{B} - \mathbf{Z}_k\mathbf{H} - \mathbf{H}^\top\mathbf{Z}_k + \mathbf{Z}_k.
\end{aligned}
$$
Let us mention that $\mathbf{H}(h^k - v) = h^k - v$ and $(h^k - v)^\top \mathbf{H}^\top = (h^k - v)^\top$ as both vectors $h^k$ and $v$ belong to Range $\left(\mathbf{A}^\top\right)$. Therefore,
$$(h^k - v)^\top \mathbb{E}_{\mathcal{D}}\left[\mathbf{B} - \mathbf{Z}_k\mathbf{H} - \mathbf{H}^\top\mathbf{Z}_k + \mathbf{Z}_k\right](h^k - v) = (h^k - v)^\top \left(\mathbf{B} - \mathbb{E}_{\mathcal{D}}[\mathbf{Z}_k]\right)(h^k - v).$$

It remains to consider
$$\mathbb{E}_{\mathcal{D}}\left[\mathbf{Z}_k\mathbf{B}^{-1}\mathbf{H}^\top\mathbf{B}\mathbf{H}\mathbf{B}^{-1}\mathbf{Z}_k\right] = \mathbb{E}_{\mathcal{D}}\left[\mathbf{Z}_k\mathbf{H}\mathbf{B}^{-1}\mathbf{B}\mathbf{H}\mathbf{B}^{-1}\mathbf{Z}_k\right] = \mathbb{E}_{\mathcal{D}}\left[\mathbf{Z}_k\right].$$
We, thereby, have derived
$$
\begin{aligned}
\mathbb{E}_{\mathcal{D}}\left[\|h^{k+1} - v\|_{\mathbf{B}}^2\right] &= (h^k - v)^\top \left(\mathbf{B} - \mathbb{E}_{\mathcal{D}}[\mathbf{Z}_k]\right)(h^k - v) \\
&\quad + (\nabla f(x^k) - v)^\top \mathbb{E}_{\mathcal{D}}\left[\mathbf{Z}_k\mathbf{B}^{-1}\mathbf{Z}_k\right](\nabla f(x^k) - v) \\
&= \|h^k - v\|_{\mathbf{B} - \mathbb{E}_{\mathcal{D}}[\mathbf{Z}_k]}^2 + \|\nabla f(x^k) - v\|_{\mathbb{E}_{\mathcal{D}}[\mathbf{Z}]}^2.
\end{aligned}
$$
$\square$

**Lemma D.5.** *Suppose $h^k \in Range\left(\mathbf{A}^\top\right)$ and $g^k$ is defined by (38). Then*

$$\mathbb{E}_{\mathcal{D}}\left[\|g^k - v\|_{\mathbf{B}}^2\right] \leq \|h^k - v\|_{\mathbf{C}-\mathbf{B}}^2 + \|\nabla f(x^k) - v\|_{\mathbf{C}}^2 \tag{44}$$

*for any $v \in Range\left(\mathbf{A}^\top\right)$, where*

$$\mathbf{C} \overset{def}{=} \mathbb{E}_{\mathcal{D}}\left[\theta^2 \mathbf{Z}\right]. \tag{45}$$

**Proof:** Writing $g^k - v = a + b$, where $a \overset{def}{=} (\mathbf{I} - \theta_k \mathbf{H}\mathbf{B}^{-1}\mathbf{Z}_k)(h^k - v)$ and $b \overset{def}{=} \theta_k \mathbf{H}\mathbf{B}^{-1}\mathbf{Z}_k(\nabla f(x^k) - v)$, we get $\|g^k\|_{\mathbf{B}}^2 \leq 2(\|a\|_{\mathbf{B}}^2 + \|b\|_{\mathbf{B}}^2)$. By definition of $\theta_k$,

$$\begin{aligned}
\mathbb{E}_{\mathcal{D}}\left[\|a\|_{\mathbf{B}}^2\right] &= \mathbb{E}_{\mathcal{D}}\left[\|\left(\mathbf{I} - \theta_k \mathbf{H}\mathbf{B}^{-1}\mathbf{Z}_k\right)(h^k - v)\|_{\mathbf{B}}^2\right] \\
&= (h^k - v)^\top \mathbb{E}_{\mathcal{D}}\left[\left(\mathbf{I} - \theta_k \mathbf{Z}_k \mathbf{B}^{-1}\mathbf{H}\right)\mathbf{B}\left(\mathbf{I} - \theta_k \mathbf{H}\mathbf{B}^{-1}\mathbf{Z}_k\right)\right](h^k - v) \\
&= (h^k - v)^\top \mathbb{E}_{\mathcal{D}}\left[\left(\mathbf{B} - \theta_k \mathbf{Z}_k \mathbf{B}^{-1}\mathbf{H}\mathbf{B} - \mathbf{B}\theta_k \mathbf{H}\mathbf{B}^{-1}\mathbf{Z}_k + \theta_k^2 \mathbf{Z}_k \mathbf{B}^{-1}\mathbf{H}\mathbf{B}\mathbf{H}\mathbf{B}^{-1}\mathbf{Z}_k\right)\right](h^k - v).
\end{aligned}$$

According to Lemma D.3, $\mathbf{H}\mathbf{B}^{-1} = \mathbf{B}^{-1}\mathbf{H}$ and $\mathbf{Z}_k\mathbf{H}\mathbf{B}^{-1}\mathbf{Z}_k = \mathbf{Z}_k$, so

$$\begin{aligned}
\mathbb{E}_{\mathcal{D}}\left[\|a\|_{\mathbf{B}}^2\right] &= (h^k - v)^\top \mathbb{E}_{\mathcal{D}}\left[\left(\mathbf{B} - \theta_k \mathbf{Z}_k \mathbf{H} - \theta_k \mathbf{H}^\top \mathbf{Z}_k + \theta_k^2 \mathbf{Z}_k\right)\right](h^k - v) \\
&= \|h^k - v\|_{\mathbb{E}_{\mathcal{D}}[\theta^2 \mathbf{Z}]-\mathbf{B}}^2,
\end{aligned}$$

where in the last step we used the assumption that $h^k$ and $v$ are from $Range\left(\mathbf{A}^\top\right)$ and $\mathbf{H}$ is the projector operator onto $Range\left(\mathbf{A}^\top\right)$.

Similarly, the second term in the upper bound on $g^k$ can be rewritten as

$$\begin{aligned}
\mathbb{E}_{\mathcal{D}}\left[\|b\|_{\mathbf{B}}^2\right] &= \mathbb{E}_{\mathcal{D}}\left[\|\theta_k \mathbf{H}\mathbf{B}^{-1}\mathbf{Z}_k(\nabla f(x^k) - v)\|_{\mathbf{B}}^2\right] \\
&= (\nabla f(x^k) - v)^\top \mathbb{E}_{\mathcal{D}}\left[\theta_k^2 \mathbf{Z}_k \mathbf{B}^{-1}\mathbf{H}^\top \mathbf{B}\mathbf{H}\mathbf{B}^{-1}\mathbf{Z}_k\right](\nabla f(x^k) - v) \\
&= \|\nabla f(x^k) - v\|_{\mathbb{E}_{\mathcal{D}}[\theta_k^2 \mathbf{Z}_k]}^2.
\end{aligned}$$

Combining the pieces, we get the claim. $\square$

## D.3 Main result

The main result of this section is:

**Theorem D.6.** *Assume that $f$ is $\mathbf{Q}$–smooth, $\mu$–strongly convex, and that $\alpha > 0$ is such that*

$$\alpha\left(2(\mathbf{C} - \mathbf{B}) + \sigma\mu\mathbf{B}\right) \leq \sigma\mathbb{E}_{\mathcal{D}}\left[\mathbf{Z}\right], \qquad \alpha\mathbf{C} \leq \frac{1}{2}\left(\mathbf{Q} - \sigma\mathbb{E}_{\mathcal{D}}\left[\mathbf{Z}\right]\right). \tag{46}$$

*If we define $\Phi^k \overset{def}{=} \|x^k - x^*\|_{\mathbf{B}}^2 + \sigma\alpha\|h^k - \nabla f(x^k)\|_{\mathbf{B}}^2$, then $\mathbb{E}\left[\Phi^k\right] \leq (1 - \alpha\mu)^k \Phi^0$.*

**Proof:** Having established Lemmas D.3, D.4 and D.5, the proof follows the same steps as the proof of Theorem 3.3. $\square$

## D.4 Optimal choice of $\mathbf{B}$ and $\mathbf{S}_k$

Let us now slightly change the value of $\theta_k$ that we use in the algorithm. Instead of seeking for $\theta_k$ giving $\mathbb{E}_{\mathcal{D}}\left[\theta_k \mathbf{Z}_k\right] = \mathbf{B}$, we will use the one that gives $\mathbb{E}_{\mathcal{D}}\left[\theta_k \mathbf{Z}_k\right] = \mathbf{B}\mathbf{H}$. This will steal lead to $\mathbb{E}_{\mathcal{D}}\left[g^k\right] = \nabla f(x^k)$ and, if $f$ is strongly-convex, we can still show the convergence rate of Theorem D.6. Although the strong convexity assumption is simplistic, the new idea results in a surprising finding.

Let $a_1, \ldots, a_m$ be the columns of $\mathbf{A}^\top$ and $\mathbf{U} \in \mathbb{R}^{d \times n}$ be a matrix that transforms these columns into an orthogonal basis of $d \overset{def}{=} Rank(\mathbf{A})$ vectors. Set $\mathbf{B} = \mathbf{U}^\top \mathbf{U}$. Then, $\langle a_i, a_j \rangle_{\mathbf{B}} = 0$ for any $i \neq j$. Assume for simplicity, that $\|a_i\|_{\mathbf{B}} \neq 0$ for $i \leq d$ and $\|a_i\|_{\mathbf{B}} = 0$ for $i > d$. This is always

true up to permutation of $a_1, \ldots, a_m$. Choose also $\mathbf{S}_k \in \mathbb{R}^n$ equal to $\xi_i \overset{\text{def}}{=} \frac{\mathbf{B}a_i}{\|a_i\|_{\mathbf{B}}}$ with $i$ sampled with probability $p_i > 0$, and $\theta_k = p_i^{-1}$. Clearly, one has

$$\mathbb{E}_{\mathcal{D}} [\theta_k \mathbf{Z}_k] = \sum_{i=1}^{d} p_i p_i^{-1} \xi_i (\xi_i^\top \mathbf{H} \mathbf{B}^{-1} \xi_i)^\dagger \xi_i^\top = \sum_{i=1}^{d} \xi_i \|a_i\|_{\mathbf{B}}^2 (a_i^\top \mathbf{B} \mathbf{H} \mathbf{B}^{-1} \mathbf{B} a_i)^\dagger \xi_i^\top.$$

Since $a_i$ lies in Range $(\mathbf{A}^\top)$, we have $\mathbf{H}a_i = a_i$, which gives

$$\mathbb{E}_{\mathcal{D}} [\theta_k \mathbf{Z}_k] = \sum_{i=1}^{d} \xi_i \|a_i\|_{\mathbf{B}}^2 (a_i^\top \mathbf{B} a_i)^\dagger \xi_i^\top = \sum_{i=1}^{d} \xi_i \xi_i^\top. \tag{47}$$

By definition of $\mathbf{B}$,

$$(\mathbf{A}\mathbf{B}\mathbf{A}^\top)^\dagger = (\text{diag}(\|a_i\|_{\mathbf{B}}^2))^\dagger = \sum_{i=1}^{d} \|a_i\|_{\mathbf{B}}^{-2} e_i e_i^\top.$$

Thus,

$$\mathbf{B}\mathbf{H} = \mathbf{B}\mathbf{A}^\top(\mathbf{A}\mathbf{B}\mathbf{A}^\top)\mathbf{A}\mathbf{B} = \sum_{i=1}^{d} \frac{(\mathbf{B}a_i)^\top \mathbf{B}a_i}{\|a_i\|_{\mathbf{B}}^2} = \mathbb{E}_{\mathcal{D}} [\theta_k \mathbf{Z}_k],$$

so we have achieved our goal. Note that if $h^0 \in$ Range $(\mathbf{A}^\top)$, we have $h^k \in$ Range $(\mathbf{A}^\top)$ even without implicitly enforcing it in (33). Therefore, the method can be seen as *SEGA with a smart choice of both sketches and metric* in which we project.

To show how the choice of $\mathbf{B}$ and of the sketches provided above improves the rate, let us take a closer look at the conditions of Theorem D.6. We have

$$\mathbf{C} \overset{(45)}{=} \mathbb{E}_{\mathcal{D}} [\theta^2 \mathbf{Z}] \overset{(47)}{=} \sum_{i=1}^{d} p_i p_i^{-2} \xi_i \xi_i^\top = \sum_{i=1}^{d} p_i^{-1} \xi_i \xi_i^\top.$$

If we assume that $\sigma \le 2/\mu$, then the first bound on $\alpha$ simplifies to

$$\alpha(2(\mathbf{C} - \mathbf{B}) + \sigma \mu \mathbf{B}) \le 2\alpha \mathbf{C} \le \sigma \mathbb{E}_{\mathcal{D}} [\mathbf{Z}] = \sigma \sum_{i=1}^{d} p_i \xi_i \xi_i^\top,$$

where the second part needs to be verified by choosing $\alpha$ to be small enough. For this it is sufficient to take $\alpha \le \sigma \max p_i^{-2}$ as every summand $\xi_i \xi_i^\top$ in the expression for $\mathbf{C}$ is positive definite. As for the second condition, it is enough to choose $\sigma \le \frac{\lambda_{\max}(\mathbf{Q})}{2\lambda_{\min}(\mathbb{E}_{\mathcal{D}}[\mathbf{Z}])}$ and $\alpha \le \frac{\lambda_{\max}(\mathbf{Q})}{4\lambda_{\min}(\mathbf{C})}$. Note that $\xi_i \xi_i^\top \le \|\xi_i\|_2^2 \mathbf{I}$, so for uniform sampling with $p_i = \frac{1}{d}$ and uniform $\mathbf{Q}$–smoothness with $\mathbf{Q} = \frac{1}{L}\mathbf{I}$ we get the following condition on $\alpha$:

$$\alpha \le \min\left\{\frac{\sigma}{d^2}, \frac{1}{4Ld \max_i \|\xi_i\|_2^2}\right\}.$$

In particular, choosing $\sigma = \min\left\{\frac{2}{\mu}, \frac{\lambda_{\max}(\mathbf{Q})}{2\lambda_{\min}(\mathbb{E}_{\mathcal{D}}[\mathbf{Z}])}\right\} = \min\left\{\frac{2}{\mu}, \frac{d}{2L \max_i \|\xi_i\|_2^2}\right\}$, we get the requirement

$$\alpha \le \min\left\{\frac{2}{\mu d^2}, \frac{1}{4Ld \max_i \|\xi_i\|_2^2}\right\}.$$

Typically, $d \ll \frac{1}{\mu}$, so the leading term in the maximum above is the second one and we get $\mathcal{O}\left(\frac{1}{d}\right)$ requirement instead of previous $\mathcal{O}\left(\frac{1}{n}\right)$.

## D.5 The conclusion of subspace `SEGA`

Let us recall that $g^k = h^k + \theta_k \mathbf{B}^{-1} \mathbf{Z}_k (\nabla f(x^k) - h^k)$. A careful examination shows that when we reduce $\theta_k$ from $\mathcal{O}(n)$ to $\mathcal{O}(d)$, we put more trust in the value of $h^k$ with the benefit of reducing the variance of $g^k$. This insight points out that a practical implementation of the algorithm may exploit the fact that $h^k$ learns the gradient of $f$ by using smaller $\theta_k$.

It is also worth noting that `SEGA` is a stationary point algorithm regardless of the value of $\theta_k$. Indeed, if one has $x^k = x^*$ and $h^k = \nabla f(x^*)$, then $g^k = \nabla f(x^*)$ for any $\theta_k$. Therefore, once we get a reasonable $h^k$, it is well grounded to choose $g^k$ to be closer to $h^k$. This argument is also supported by our experiments.

Finally, the ability to take bigger stepsizes is also of high interest. One can think of extending other methods in this direction, especially if interested in applications with a small rank of matrix $\mathbf{A}$.

## E  Simplified Analysis of `SEGA` 1

In this section we consider the setup from Example 2.1 with $\mathbf{B} = \mathbf{I}$ uniform probabilities: $p_i = 1/n$ for all $i$ and proximal term $R = 0$. We now state the main complexity result.

**Theorem E.1.** *Let* $\mathbf{B} = \mathbf{I}$ *and choose* $\mathcal{D}$ *to be the uniform distribution over unit basis vectors in* $\mathbb{R}^n$. *Choose* $\sigma > 0$ *and define*

$$\Phi^k \overset{def}{=} \|x^k - x^*\|_2^2 + \sigma\alpha\|h^k\|_2^2,$$

*where* $\{x^k, h^k\}_{k \geq 0}$ *are the iterates of the gradient sketch method. If the stepsize satisfies*

$$0 < \alpha \leq \min\left\{\frac{1 - \frac{L\sigma}{n}}{2Ln}, \frac{1}{n\left(\mu + \frac{2(n-1)}{\sigma}\right)}\right\}, \tag{48}$$

*then* $\mathbb{E}_{\mathcal{D}}\left[\Phi^{k+1}\right] \leq (1 - \alpha\mu)\Phi^k$. *This means that*

$$k \geq \frac{1}{\alpha\mu}\log\frac{1}{\epsilon} \quad \Rightarrow \quad \mathbb{E}\left[\Phi^k\right] \leq \epsilon\Phi^0.$$

In particular, if we let $\sigma = \frac{n}{2L}$, then $\alpha = \frac{1}{(4L+\mu)n}$ satisfies (48), and we have the iteration complexity

$$n\left(4 + \frac{1}{\kappa}\right)\kappa\log\frac{1}{\epsilon} = \tilde{\mathcal{O}}(n\kappa),$$

where $\kappa \overset{def}{=} \frac{L}{\mu}$ is the condition number.

This is the same complexity as `NSync` [41] under the same assumptions on $f$. `NSync` also needs just access to partial derivatives. However, `NSync` uses variable stepsizes, while `SEGA` can do the same with *fixed* stepsizes. This is because `SEGA` *learns* the direction $g^k$ using past information.

### E.1  Technical Lemmas

Since $f$ is $L$–smooth, we have

$$\|\nabla f(x^k)\|_2^2 \leq 2L(f(x^k) - f(x^*)). \tag{49}$$

On the other hand, by $\mu$–strong convexity of $f$ we have

$$f(x^*) \geq f(x^k) + \langle\nabla f(x^k), x^* - x^k\rangle + \frac{\mu}{2}\|x^* - x^k\|_2^2. \tag{50}$$

**Lemma E.2.** *The variance of $g^k$ as an estimator of $\nabla f(x^k)$ can be bounded as follows:*

$$\mathbb{E}_{\mathcal{D}}\left[\|g^k\|_2^2\right] \leq 4Ln(f(x^k) - f(x^*)) + 2(n-1)\|h^k\|_2^2. \tag{51}$$

**Proof:** In view of (9), we first write

$$g^k = \underbrace{h^k - \frac{1}{p_i}e_i^\top h^k e_i}_{a} + \underbrace{\frac{1}{p_i}e_i^\top \nabla f(x^k)e_i}_{b},$$

and note that $p_i = 1/n$ for all $i$. Let us bound the expectation of each term individually. The first term is equal to

$$
\begin{aligned}
\mathbb{E}_{\mathcal{D}}\left[\|a\|_2^2\right] &= \mathbb{E}_{\mathcal{D}}\left[\|h^k - ne_i^\top h^k e_i\|_2^2\right] \\
&= \mathbb{E}_{\mathcal{D}}\left[\|\left(\mathbf{I} - ne_i e_i^\top\right)h^k\|_2^2\right] \\
&= (h^k)^\top \mathbb{E}_{\mathcal{D}}\left[\left(\mathbf{I} - ne_i e_i^\top\right)^\top \left(\mathbf{I} - ne_i e_i^\top\right)\right] h^k \\
&= (n-1)\|h^k\|_2^2.
\end{aligned}
$$

The second term can be bounded as

$$
\begin{aligned}
\mathbb{E}_{\mathcal{D}}\left[\|b\|_2^2\right] &= \mathbb{E}_{\mathcal{D}}\left[\|ne_i^\top \nabla f(x^k)e_i\|_2^2\right] \\
&= n^2 \sum_{i=1}^{n} \frac{1}{n}(e_i^\top \nabla f(x^k))^2 \\
&= n\|\nabla f(x^k)\|_2^2 \\
&= n\|\nabla f(x^k) - \nabla f(x^*)\|_2^2 \\
&\overset{(49)}{\leq} 2Ln(f(x^k) - f(x^*)),
\end{aligned}
$$

where in the last step we used $L$–smoothness of $f$. It remains to combine the two bounds.

**Lemma E.3.** *For all $k$ we have*

$$\mathbb{E}_{\mathcal{D}}\left[\|h^{k+1}\|_2^2\right] = \left(1 - \frac{1}{n}\right)\|h^k\|_2^2 + \frac{1}{n}\|\nabla f(x^k)\|_2^2. \tag{52}$$

**Proof:** We have

$$
\begin{aligned}
\mathbb{E}_{\mathcal{D}}\left[\|h^{k+1}\|_2^2\right] &\overset{(8)}{=} \mathbb{E}_{\mathcal{D}}\left[\|h^k + e_{i_k}^\top(\nabla f(x^k) - h^k)e_{i_k}\|_2^2\right] \\
&= \mathbb{E}_{\mathcal{D}}\left[\|\left(\mathbf{I} - e_{i_k}e_{i_k}^\top\right)h^k + e_{i_k}e_{i_k}^\top \nabla f(x^k)\|_2^2\right] \\
&= \mathbb{E}_{\mathcal{D}}\left[\|\left(\mathbf{I} - e_{i_k}e_{i_k}^\top\right)h^k\|_2^2\right] + \mathbb{E}_{\mathcal{D}}\left[\|e_{i_k}e_{i_k}^\top \nabla f(x^k)\|_2^2\right] \\
&= (h^k)^\top \mathbb{E}_{\mathcal{D}}\left[\left(\mathbf{I} - e_{i_k}e_{i_k}^\top\right)^\top \left(\mathbf{I} - e_{i_k}e_{i_k}^\top\right)\right]h^k (\nabla f(x^k))^\top \mathbb{E}_{\mathcal{D}}\left[(e_{i_k}e_{i_k}^\top)^\top e_{i_k}e_{i_k}^\top\right]\nabla f(x^k) \\
&= (h^k)^\top \mathbb{E}_{\mathcal{D}}\left[\mathbf{I} - e_{i_k}e_{i_k}^\top\right]h^k + (\nabla f(x^k))^\top \mathbb{E}_{\mathcal{D}}\left[e_{i_k}e_{i_k}^\top\right]\nabla f(x^k) \\
&= \left(1 - \frac{1}{n}\right)\|h^k\|_2^2 + \frac{1}{n}\|\nabla f(x^k)\|_2^2.
\end{aligned}
$$

## E.2 Proof of Theorem E.1

We can now write

$$
\begin{aligned}
\mathbb{E}_{\mathcal{D}}\left[\|x^{k+1} - x^*\|_2^2\right] &= \mathbb{E}_{\mathcal{D}}\left[\|x^k - \alpha g^k - x^*\|_2^2\right] \\
&= \|x^k - x^*\|_2^2 + \alpha^2 \mathbb{E}_{\mathcal{D}}\left[\|g^k\|_2^2\right] - 2\alpha\langle\mathbb{E}_{\mathcal{D}}\left[g^k\right], x^k - x^*\rangle \\
&\overset{(7)}{=} \|x^k - x^*\|_2^2 + \alpha^2 \mathbb{E}_{\mathcal{D}}\left[\|g^k\|_2^2\right] - 2\alpha\langle\nabla f(x^k), x^k - x^*\rangle \\
&\overset{(50)}{\leq} (1 - \alpha\mu)\|x^k - x^*\|_2^2 + \alpha^2 \mathbb{E}_{\mathcal{D}}\left[\|g^k\|_2^2\right] - 2\alpha(f(x^k) - f(x^*)).
\end{aligned}
$$

Using Lemma E.2, we can further estimate

$$
\begin{aligned}
\mathbb{E}_{\mathcal{D}}\left[\|x^{k+1} - x^*\|_2^2\right] \leq{} & (1 - \alpha\mu)\|x^k - x^*\|_2^2 \\
& + 2\alpha(2Ln\alpha - 1)(f(x^k) - f(x^*)) + 2(n-1)\alpha^2\|h^k\|_2^2.
\end{aligned}
$$

Let us now add $\sigma\alpha\mathbb{E}_{\mathcal{D}}\left[\|h^{k+1}\|_2^2\right]$ to both sides of the last inequality. Recalling the definition of the Lyapunov function, and applying Lemma B.3, we get

$$
\begin{aligned}
\mathbb{E}_{\mathcal{D}}\left[\Phi^{k+1}\right] \leq{} & (1 - \alpha\mu)\|x^k - x^*\|_2^2 + 2\alpha(2Ln\alpha - 1)(f(x^k) - f(x^*)) + 2(n-1)\alpha^2\|h^k\|_2^2 \\
& + \sigma\alpha\left(1 - \frac{1}{n}\right)\|h^k\|_2^2 + \frac{\sigma\alpha}{n}\|\nabla f(x^k)\|_2^2 \\
\overset{(49)}{\leq}{} & (1 - \alpha\mu)\|x^k - x^*\|_2^2 + 2\alpha\underbrace{\left(2Ln\alpha + \frac{L\sigma}{n} - 1\right)}_{\text{I}}(f(x^k) - f(x^*)) \\
& + \underbrace{\left(1 - \frac{1}{n} + \frac{2(n-1)\alpha}{\sigma}\right)}_{\text{II}}\sigma\alpha\|h^k\|_2^2.
\end{aligned}
$$

Let us choose $\alpha$ so that I $\leq 0$ and II $\leq 1 - \alpha\mu$. This leads to the bound (48). For any $\alpha > 0$ satisfying this bound we therefore have $\mathbb{E}_{\mathcal{D}}\left[\Phi^{k+1}\right] \leq (1 - \alpha\mu)\Phi^k$, as desired. Lastly, as we have freedom to choose $\sigma$, let us pick it so as to maximize the upper bound on the stepsize.

## F  Simplified Analysis of SEGA II

In this section we consider the setup from Example 2.1 with arbitrary non-uniform probabilities: $p_i > 0$ for all $i$ and proximal term $R = 0$. We provide a simplified analysis of SEGA in this scenario. However, we will do this under slightly different assumptions. In particular, we shall assume that smoothness and strong convexity of $f$ are measured with respect to the same norm.

In this setup, as we shall see, uniform probabilities are optimal. That is, uniform probabilities are identical to the importance sampling probabilities. We note that this would be the case even for standard coordinate descent under these assumptions, as follows from the results in [41].

Let $\mathbf{G} = \mathrm{Diag}(g_1, \ldots, g_n) \succ 0$ and assume that

$$
\|\nabla f(x) - \nabla f(y)\|_{\mathbf{G}^{-1}} \leq L\|x - y\|_{\mathbf{G}}
$$

and[7]

$$
f(x) \geq f(y) + \langle \nabla f(y), x - y\rangle + \frac{\mu}{2}\|x - y\|_{\mathbf{G}}^2
$$

for all $x, y \in \mathbb{R}^n$. These two assumptions combined lead to the following inequalities:

$$
f(y) + \langle \nabla f(y), x - y\rangle + \frac{\mu}{2}\|x - y\|_{\mathbf{G}}^2 \leq f(x) \leq f(y) + \langle \nabla f(y), x - y\rangle + \frac{L}{2}\|x - y\|_{\mathbf{G}}^2.
$$

We define $g^k$ as before, but change the method to:

$$
\boxed{x^{k+1} = x^k - \alpha\mathbf{G}^{-1}g^k} \tag{53}
$$

We now state the main complexity result.

**Theorem F.1.** *Choose $\sigma > 0$ and define $\Phi^k \stackrel{\text{def}}{=} \|x^k - x^*\|_{\mathbf{G}}^2 + \sigma\alpha\|h^k\|_{\mathrm{Diag}\left(\frac{1}{g_i p_i}\right)}^2$, where $\{x^k, h^k\}_{k \geq 0}$ are the iterates of the gradient sketch method. If the stepsize satisfies*

$$0 < \alpha \leq \min_i \left\{ p_i \left( \frac{1}{\mu + L} - \frac{\sigma}{2} \right), \frac{p_i}{\frac{2}{\sigma}(1 - p_i) + \frac{2L\mu}{\mu+L}} \right\}, \tag{54}$$

*then $\mathbb{E}_{\mathcal{D}}\left[ \Phi^{k+1} \right] \leq \left( 1 - \alpha\mu\frac{2L}{\mu+L} \right) \Phi^k$. This means that*

$$k \geq \frac{L + \mu}{2\alpha L\mu} \log\frac{1}{\epsilon} \quad \Rightarrow \quad \mathbb{E}\left[ \Phi^k \right] \leq \epsilon\Phi^0.$$

*In particular, if we choose $g_i = 1$ and $p_i = \frac{1}{n}$ for all $i$, then if we set $\sigma = \frac{1}{2L}$, we can choose stepsize $\alpha = \frac{3L-\mu}{4Ln(L+\mu)}$, and obtain the rate $\frac{2L+2\mu}{3L-\mu}n \left( \frac{L}{\mu} + 1 \right) \log\frac{1}{\epsilon} \leq 2n \left( \frac{L}{\mu} + 1 \right) \log\frac{1}{\epsilon}$.*

## F.1 Two lemmas

**Lemma F.2.** *Let $d_1, \ldots, d_n > 0$. The variance of $g^k$ as an estimator of $\nabla f(x^k)$ can be bounded as follows:*

$$\mathbb{E}_{\mathcal{D}}\left[ \|g^k\|_{\mathrm{Diag}(d_i)}^2 \right] \leq 2\|h^k\|_{\mathrm{Diag}\left(d_i\frac{1-p_i}{p_i}\right)}^2 + 2\|\nabla f(x^k)\|_{\mathrm{Diag}\left(\frac{d_i}{p_i}\right)}^2. \tag{55}$$

**Proof:** In view of (9), we first write

$$g^k = \underbrace{h^k - \frac{1}{p_i}e_i^\top h^k e_i}_{a} + \underbrace{\frac{1}{p_i}e_i^\top \nabla f(x^k)e_i}_{b}.$$

Let us bound the expectation of each term individually. The first term is equal to

$$
\begin{aligned}
\mathbb{E}_{\mathcal{D}}\left[ \|a\|_{\mathbf{G}^{-1}}^2 \right] &= \mathbb{E}_{\mathcal{D}}\left[ \left\| h^k - \frac{1}{p_i}e_i^\top h^k e_i \right\|_{\mathrm{Diag}(d_i)}^2 \right] \\
&= \mathbb{E}_{\mathcal{D}}\left[ \left\| \left( \mathbf{I} - \frac{1}{p_i}e_i e_i^\top \right) h^k \right\|_{\mathrm{Diag}(d_i)}^2 \right] \\
&= (h^k)^\top \mathbb{E}_{\mathcal{D}}\left[ \left( \mathbf{I} - \frac{1}{p_i}e_i e_i^\top \right)^\top \mathrm{Diag}(d_i) \left( \mathbf{I} - \frac{1}{p_i}e_i e_i^\top \right) \right] h^k \\
&= (h^k)^\top \mathbb{E}_{\mathcal{D}}\left[ \left( \mathrm{Diag}(d_i) - \frac{2d_i}{p_i}e_i e_i^\top + \frac{d_i}{p_i^2}e_i e_i^\top \right) \right] h^k \\
&= \sum_{i=1}^n d_i \left( \frac{1}{p_i} - 1 \right)(h_i^k)^2.
\end{aligned}
$$

The second term can be bounded as

$$\mathbb{E}_{\mathcal{D}}\left[ \|b\|_{\mathrm{Diag}(d_i)}^2 \right] = \mathbb{E}_{\mathcal{D}}\left[ \left\| \frac{1}{p_i}e_i^\top \nabla f(x^k)e_i \right\|_{\mathrm{Diag}(d_i)}^2 \right] = \sum_{i=1}^n \frac{d_i}{p_i}(e_i^\top \nabla f(x^k))^2.$$

It remains to combine the two bounds. $\qquad\square$

**Lemma F.3.** *For all $v \in \mathbb{R}^n$ and $d_1, \ldots, d_n > 0$ we have*

$$\mathbb{E}_{\mathcal{D}}\left[ \|h^{k+1} - v\|_{\mathrm{Diag}(d_i)}^2 \right] = \|h^k - v\|_{\mathrm{Diag}(d_i(1-p_i))}^2 + \|\nabla f(x^k) - v\|_{\mathrm{Diag}(d_i p_i)}^2. \tag{56}$$

**Proof:** We have

$$\mathbb{E}_{\mathcal{D}}\left[\|h^{k+1}-v\|^2_{\mathrm{Diag}(d_i)}\right] \overset{(8)}{=} \mathbb{E}_{\mathcal{D}}\left[\|h^k + e_i^\top(\nabla f(x^k)-h^k)e_i - v\|^2_{\mathrm{Diag}(d_i)}\right]$$

$$= \mathbb{E}_{\mathcal{D}}\left[\|\left(\mathbf{I}-e_ie_i^\top\right)(h^k-v) + e_ie_i^\top(\nabla f(x^k)-v)\|^2_{\mathrm{Diag}(d_i)}\right]$$

$$= \mathbb{E}_{\mathcal{D}}\left[\|\left(\mathbf{I}-e_ie_i^\top\right)(h^k-v)\|^2_{\mathrm{Diag}(d_i)}\right] + \mathbb{E}_{\mathcal{D}}\left[\|e_ie_i^\top(\nabla f(x^k)-v)\|^2_{\mathrm{Diag}(d_i)}\right]$$

$$= (h^k-v)^\top \mathbb{E}_{\mathcal{D}}\left[\left(\mathbf{I}-e_ie_i^\top\right)^\top \mathrm{Diag}(d_i)\left(\mathbf{I}-e_ie_i^\top\right)\right](h^k-v)$$
$$+ (\nabla f(x^k)-v)^\top \mathbb{E}_{\mathcal{D}}\left[(e_ie_i^\top)^\top \mathrm{Diag}(d_i)e_ie_i^\top\right](\nabla f(x^k)-v)$$

$$= (h^k-v)^\top \mathbb{E}_{\mathcal{D}}\left[\mathrm{Diag}(d_i) - d_ie_ie_i^\top\right](h^k-v)$$
$$+ (\nabla f(x^k)-v)^\top \mathbb{E}_{\mathcal{D}}\left[d_ie_ie_i^\top\right](\nabla f(x^k)-v)$$

$$= \|h^k-v\|^2_{\mathrm{Diag}(d_i(1-p_i))} + \|\nabla f(x^k)-v\|^2_{\mathrm{Diag}(d_ip_i)}.$$

$\square$

## F.2 Proof of Theorem F.1

**Proof:** Since $f$ is $L$–smooth and $\mu$–strongly convex, we have the inequality

$$\langle \nabla f(x) - \nabla f(y), x-y\rangle \geq \frac{\mu L}{\mu+L}\|x-y\|^2_{\mathbf{G}} + \frac{1}{\mu+L}\|\nabla f(x)-\nabla f(y)\|^2_{\mathbf{G}^{-1}}.$$

In particular, we will use it for $x = x^k$ and $y = x^*$:

$$\langle \nabla f(x^k), x^* - x^k\rangle \leq -\frac{\mu L}{\mu+L}\|x-x^*\|^2_{\mathbf{G}} - \frac{1}{\mu+L}\|\nabla f(x^k)\|^2_{\mathbf{G}^{-1}}. \qquad (57)$$

We can now write

$$\mathbb{E}_{\mathcal{D}}\left[\|x^{k+1}-x^*\|^2_{\mathbf{G}}\right] \overset{(53)}{=} \mathbb{E}_{\mathcal{D}}\left[\|x^k - \alpha\mathbf{G}^{-1}g^k - x^*\|^2_{\mathbf{G}}\right]$$

$$= \|x^k-x^*\|^2_{\mathbf{G}} + \alpha^2\mathbb{E}_{\mathcal{D}}\left[\|\mathbf{G}^{-1}g^k\|^2_{\mathbf{G}}\right] - 2\alpha\langle\mathbb{E}_{\mathcal{D}}\left[g^k\right], x^k-x^*\rangle$$

$$\overset{(7)}{=} \|x^k-x^*\|^2_{\mathbf{G}} + \alpha^2\mathbb{E}_{\mathcal{D}}\left[\|g^k\|^2_{\mathbf{G}^{-1}}\right] + 2\alpha\langle\nabla f(x^k), x^*-x^k\rangle$$

$$\overset{(57)}{\leq} \left(1-\alpha\mu\frac{2L}{\mu+L}\right)\|x^k-x^*\|^2_{\mathbf{G}} + \alpha^2\mathbb{E}_{\mathcal{D}}\left[\|g^k\|^2_{\mathbf{G}^{-1}}\right] - \frac{2\alpha}{\mu+L}\|\nabla f(x^k)\|^2_{\mathbf{G}^{-1}}.$$

Using Lemma F.2 to bound $\mathbb{E}_{\mathcal{D}}\left[\|g^k\|^2_{\mathbf{G}^{-1}}\right]$, we can further estimate

$$\mathbb{E}_{\mathcal{D}}\left[\|x^{k+1}-x^*\|^2_{\mathbf{G}}\right] \leq \left(1-\alpha\mu\frac{2L}{\mu+L}\right)\|x^k-x^*\|^2_{\mathbf{G}} + 2\alpha^2\|\nabla f(x^k)\|^2_{\mathrm{Diag}\left(\frac{1}{p_ig_i}\right)}$$

$$- \frac{2\alpha}{\mu+L}\|\nabla f(x^k)\|^2_{\mathbf{G}^{-1}} + 2\alpha^2\|h^k\|^2_{\mathrm{Diag}\left(\frac{1-p_i}{p_ig_i}\right)}.$$

Let us now add $\sigma\alpha\mathbb{E}_{\mathcal{D}}\left[\|h^{k+1}\|^2_{\mathrm{Diag}\left(\frac{1}{g_ip_i}\right)}\right]$ to both sides of the last inequality. Recalling the definition of the Lyapunov function, and applying Lemma F.3 with $v = 0$ and $d_i = \frac{1}{g_ip_i}$, we get

$$\mathbb{E}_{\mathcal{D}}\left[\Phi^{k+1}\right] \leq \left(1-\alpha\mu\frac{2L}{\mu+L}\right)\|x^k-x^*\|^2_{\mathbf{G}} + 2\alpha^2\|\nabla f(x^k)\|^2_{\mathrm{Diag}\left(\frac{1}{p_ig_i}\right)} + \left(\alpha\sigma - \frac{2\alpha}{\mu+L}\right)\|\nabla f(x^k)\|^2_{\mathbf{G}^{-1}}$$

$$+ (2\alpha^2+\alpha\sigma)\|h^k\|^2_{\mathrm{Diag}\left(\frac{1-p_i}{p_ig_i}\right)}$$

$$\leq \left(1-\alpha\mu\frac{2L}{\mu+L}\right)\|x^k-x^*\|^2_{\mathbf{G}} + \sigma\alpha\|h^k\|^2_{\mathrm{Diag}\left(\left(\frac{2\alpha}{\sigma}+1\right)\frac{1-p_i}{p_ig_i}\right)}$$

$$+ \|\nabla f(x^k)\|^2_{\mathrm{Diag}\left(\frac{2\alpha^2}{p_ig_i}+\frac{\sigma\alpha}{g_i}-\frac{2\alpha}{(\mu+L)g_i}\right)}.$$

If we now choose $\alpha > 0$ such that

$$\frac{2\alpha}{p_i} + \sigma - \frac{2}{\mu + L} \leq 0, \qquad \left( \frac{2\alpha}{\sigma} + 1 \right)(1 - p_i) \leq 1 - \alpha\mu\frac{2L}{\mu + L},$$

then we get the recursion

$$\mathbb{E}_{\mathcal{D}}\left[ \Phi^{k+1} \right] \leq \left( 1 - \alpha\mu\frac{2L}{\mu+L} \right) \Phi^k \leq (1 - \alpha\mu)\Phi^k.$$

$\square$

## G   Extra Experiments

### G.1   Evolution of Iterates: Extra Plots

Here we show some additional plots similar to Figure 1, which we believe help to build intuition about how the iterates of SEGA behave. We also include plots for biasSEGA, which uses biased estimators of the gradient instead. We found that the iterates of biasSEGA often behave in a more stable way, as could be expected given the fact that they enjoy lower variance. However, we do not have any theory supporting the convergence of biasSEGA; this is left for future research.

Figure 5: Evolution of iterates of SEGA, CD and biasSEGA (updates made via $h^{k+1}$ instead of $g^k$).

Figure 6: Iterates of SEGA, CD and biasSEGA (updates made via $h^{k+1}$ instead of $g^k$). Different starting point.

Figure 7: Iterates of projected SEGA, projected CD (which do not converge) and projected biasSEGA (updates made via $h^{k+1}$ instead of $g^k$). The constraint set is represented by the shaded region.

## G.2 Experiments from Section 5 with empirically optimal stepsize

In the experiments in Section 5, we worked with quadratic functions of the form

$$f(x) \stackrel{\text{def}}{=} \frac{1}{2} x^\top \mathbf{M} x - b^\top x,$$

where $b$ is a random vector with independent entries from $\mathcal{N}(0, 1)$ and $\mathbf{M} \stackrel{\text{def}}{=} \mathbf{U} \Sigma \mathbf{U}^\top$ according to Table 2 for $\mathbf{U}$ obtained from QR decomposition of random matrix with independent entries from $\mathcal{N}(0, 1)$. For each problem, the starting point was chosen to be a vector with independent entries from $\mathcal{N}(0, 1)$.

| Type | $\Sigma$ |
|------|----------|
| 1 | Diagonal matrix with first $n/2$ components equal to 1 and the rest equal to $n$ |
| 2 | Diagonal matrix with first $n-1$ components equal to 1 and the remaining one equal to $n$ |
| 3 | Diagonal matrix with $i$–th component equal to $i$ |
| 4 | Diagonal matrix with components coming from uniform distribution over $[0, 1]$ |

Table 2: Spectrum of $\mathbf{M}$.

The results are provided in Figures 8-10. They include zeroth-order experiments and the subspace version of SEGA.

Figure 8: Counterpart to Figure 2 – convergence illustration of SEGA and PGD. The indicator "Xn" in the label stands for the setting when the cost of solving linear system is $Xn$ times higher comparing to the oracle call. Recall that a linear system is solved after each $n$ oracle calls. Empirically best stepsizes were used both PGD and SEGA.

Figure 9: Counterpart to Figure 3 – comparison of SEGA and randomized direct search for a various problems. Empirically best stepsizes were used for both methods.

## G.3 Experiment: comparison with randomized coordinate descent

In this section we numerically compare the results from Section 4 to analogous results for coordinate descent (as indicated in Table 1). We consider the ridge regression problem on LibSVM [7] data, for both primal and dual formulation. For all methods, we have chosen parameters as suggested from theory Figure 11 shows the results. We can see that in all cases, SEGA is slower to the corresponding coordinate descent method, but still is competitive. We however observe only constant times difference in terms of the speed, as suggested by Table 1.

Figure 10: Counterpart to Figure 4 – comparison of SEGA with sketches from a correct subspace versus naive SEGA. Optimal (empirically) stepsize chosen.

Figure 11: Comparison of SEGA and ASEGA with corresponding coordinate descent methods for $R = 0$.

Figure 12: Comparison of `SEGA` with `CD` on logistic regression problem with similar stepsizes.

## G.4 Experiment: large-scale logistic regression

In this experiment, we set $\mathbf{B}$ to be identity matrix and compare `CD` to `SEGA` with coordinate sketches, both with uniform sampling and with similar stepsizes. The problem considered is logistic regression with $\ell_2$ penalty:

$$\min_{x \in \mathbb{R}^n} \frac{1}{m} \sum_{i=1}^m \log\left(1 + \exp(-b_i a_i^\top x)\right) + \frac{\mu}{2}\|x\|_2^2,$$

where $a_i$ and $b_i$ are data-dependent. Clearly, this regularizer is separable, so we can easily apply both methods. The value of $\mu$ was chosen to be of order $\frac{1}{m}$ in both experiments. Here we use real-world large scale datasets from the LIBSVM [7] library, a summary can be found in Table 3. To make it clear whether `CD` and `SEGA` converge with the same speed if given similar stepsizes, we use stepsize $\frac{1}{L}$ for `CD` and $\frac{1}{dL}$ for `SEGA`. The results can be found in Figure 12.

| Dataset | $m$ | $n$ | $L$ | $\mu$ |
|---|---|---|---|---|
| Epsilon | 400000 | 2000 | 0.25 | $2.5 \cdot 10^{-5}$ |
| Covtype | 581012 | 54 | 21930585. 25 | $10^{-1}$ |

Table 3: Description of the datasets used in our logistic regression experiments. Constants $m$, $n$, $L$ and $\mu$ denote respectively the size of the training set, the number of features, the Lipschitz constant, and the value of $\ell_2$ penalty.

# H   Frequently Used Notation

| Basic | | |
|---|---|---|
| $\mathbb{E}\left[\cdot\right], \mathbb{P}\left(\cdot\right)$ | Expectation / Probability | |
| $\langle\cdot,\cdot\rangle_{\mathbf{B}}, \|\cdot\|_{\mathbf{B}}$ | Weighted inner product and norm: $\langle x,y\rangle_{\mathbf{B}} = x^{\top}\mathbf{B}y$; $\|x\|_{\mathbf{B}} = \sqrt{\langle x,x\rangle_{\mathbf{B}}}$ | |
| $e_i$ | $i$-th vector from the standard basis | |
| $\mathbf{I}$ | Identity matrix | |
| $\lambda_{\max}(\cdot), \lambda_{\min}(\cdot)$ | Maximal eigenvalue / minimal eigenvalue | |
| $f$ | Objective to be minimized over set $\mathbb{R}^n$ | (1) |
| $R$ | Regularizer | (1) |
| $x^*$ | Global optimum | |
| $L$ | Lipschitz constant for $\nabla f$ | |
| $\mathbf{Q}$ | Smoothness matrix | (10) |
| $\mathbf{M}$ | Smoothness matrix, equal to $\mathbf{Q}^{-1}$ for $\mathbf{B} = \mathbf{I}$ | (11) |
| $\mu$ | Strong convexity constant | |
| SEGA | | |
| $\mathcal{D}$ | Distribution over sketch matrices $\mathbf{S}$ | |
| $\mathbf{S}$ | Sketch matrix | (3) |
| $\mathbb{E}_{\mathcal{D}}\left[\cdot\right]$ | Expectation over the choice of $\mathbf{S}$ | |
| $b$ | Random variable such that $\mathbf{S} \in \mathbb{R}^{n\times b}$ | |
| $\zeta(\mathbf{S},x)$ | Sketched gradient at $x$ | (2) |
| $\mathbf{Z}$ | $\mathbf{S}\left(\mathbf{S}^{\top}\mathbf{B}^{-1}\mathbf{S}\right)^{\dagger}\mathbf{S}^{\top}$ | |
| $\theta$ | Random variable for which $\mathbb{E}_{\mathcal{D}}\left[\theta\mathbf{Z}\right] = \mathbf{B}$ | (5) |
| $\mathbf{C}$ | $\mathbb{E}_{\mathcal{D}}\left[\theta^2\mathbf{Z}\right]$ | Thm 3.3 |
| $h, g$ | Biased and unbiased gradient estimators | (4), (6) |
| $\alpha$ | Stepsize | |
| $\Phi$ | Lyapunov function | Thm 3.3, |
| $\sigma$ | Parameter for Lyapunov function | Thm 3.3, 4.2 |
| Extra Notation for Section 4 | | |
| $p, \mathbf{P}$ | Probability vector and matrix | |
| $v$ | vector of ESO parameters | (14) |
| $\hat{\mathbf{P}}, \hat{\mathbf{V}}$ | $\mathrm{Diag}(p), \mathrm{Diag}(v)$ | |
| $\gamma$ | $\alpha - \alpha^2 \max_i\{\frac{v_i}{p_i}\} - \sigma$ | Thm 4.2 |
| $y, z$ | Extra sequences of iterates for ASEGA | |
| $\tau, \beta$ | Parameters for ASEGA | |
| $\Psi, \Upsilon$ | Lyapunov functions | Thm 4.2, C.5 |
| $\eta(v,p)$ | $\max_i \frac{\sqrt{v_i}}{p_i}$ | |

Table 4: Summary of frequently used notation.