[Reviews · NeurIPS 2018]

Reviewer 1



In this paper, the authors propose a randomized first order optimization method (SEGA) which progressively builds a variance reduced estimate of the gradient from random linear measurements of the gradient. The proposed method (or class of methods - depending on the sketch matrix and metric used) updates the current estimate of the gradient through a sketch-and-project operation using new gradient information and the past estimate of the gradient. - The first half of the paper is very well written, easy to follow and well motivated. However, the quality of the paper deteriorates after page 6. The paper has minor typos and grammatical mistakes that can be corrected easily. — Line 29: now draw a two simple -> now draw two simple — Line 36: methods as stochastic gradient -> methods such as stochastic gradient — Line 83: For the illustration, -> To illustrate this, — Line 83: shows a the evolution -> shows the evolution — Line 87: baseline choice -> baseline choices — Line 106: no proximal setting, coordinate -> no proximal term and coordinate — Line 115: eigen-properties or matrices -> eigen-properties of matrices — Line 148: analogy -> analog (and other instances of this) — Line 152: fresult -> result — Line 158: ary -> Corollary — Line 165: in sense -> in the sense — Line 181: verify claim -> verify the claim — Line 192: a construct -> a constructed problem — Line 193: system comparing -> system compared - The Experiments section seems weak, primarily because all problems solved are constructed problems. The experiments are well though out to highlight certain algorithmic features of the method, however, several details are missing (e.g., what is the dimension n of the problems solved?), comparison with more methods would strengthen the claims made and experiments on real ML problems would highlight the merits (and limitations) of SEGA. — Section 5.1: —— The authors should mention in what settings different cost scenarios arise for solving the linear systems. —— The authors should compare the methods on problems that are not quadratic. — Section 5.2: —— How was the finite difference interval set for the gradient estimation used in SEGA? Did the authors use forward differences (n function evaluations per iteration)? ——The authors should compare against other zero order optimization methods (e.g., model based trust region method, finite difference quasi-Newton methods or Nelder-Mead). —— Again, the authors should compare the methods on problems that are not quadratic. —— In this section do the authors assume that they have access to true (non-noisy) function evaluations? — Section 5.3: —— How is beta chosen in the accelerated variant of the method? —— In practice, on real world problems, how would one chose the subspaces to use in subspaceSEGA? The authors should make a comment about this in the paper. — The authors claim that “theory supported step sizes were chosen for all experiments.” These step sizes depend on problem specific parameters such as Trace(M), minimum and maximum eigenvalues of different matrices and other quantities. How were these quantities calculated? - Other minor issues and questions: — The authors should mention clearly that it is not necessary to compute the full gradient in Algorithm 1. — “SEGA can be seen as a variance reduced coordinate descent.” Is it not the case that SEGA is more general than that? If certain sketch matrices are chosen then it is a coordinate descent type method, but with other sketches this is not the case. — “potentially more expensive iteration”: it would be helpful for the reader if the authors provided some examples. — “total complexity”: do the authors mean iteration complexity? — What is alpha_0 in Corollary 3.4? — Line 128: “Corollary 4.3 and Corollary 4.3”? — Remark 4.4: “Therefore, SEGA also works in a non-convex setting.” I believe this statement is too strong. SEGA works for a certain class of non-convex function that satisfy the PL inequality (these function have unique minimizers). — Corollary 4.5: Is the Lyapunov function used here the same as in Theorem 4.2? — Table 1: What is the cause of the constant differences in the results of CD and SEGA? Is this an artifact of the analysis? Is this realized in practice? — The authors should consider adding a Final Remarks section to summarize their contributions and to aid the reader. - I believe the generality of the algorithms the authors propose (different sketch matrices) and the special cases of algorithms that they recover, as well as the wide range of potential problem classes that can be solved by the proposed methods, makes this a paper of interest for the ML and Optimization communities. However, there are certain issues and questions with the paper (mentioned above) that should be addressed. As such, I marginally recommend this paper for publication at NIPS. I have read the author rebuttal and stand by my initial assessment of the paper.

Reviewer 2



The paper proposes a variance reduction based coordinate descent algorithm for composite optimization. The main idea is to sequentially construct an unbiased gradient estimate based on the current partial derivative and previous estimate. The proposed algorithm achieves a linear convergence rate similar to randomized coordinate descent when the problem is strongly convex. I find the idea of combining variance reduction with coordinate descent very interesting. However, the paper looks quite messy in its writing. For instance, many of the symbols/definitions are even not defined before its first appearance. Moreover, the technical results are very hard to go through due to the lack of explanation and discussion. Overall, I find the paper interesting but also a lot of room for improvement. Here are a list of points to be clarified: 1) It is assumed that the proximal operator according to an arbitrary metric B is available. However, such proximal operator is in general hard to evaluate unless the regularization function is separable. This makes the claim of non separable proximal term unfair. Moreover, is it true that other's analysis can not be extend to non separable regularizers? 2) The introduction of the metric B makes the presentation confusing. I would suggest to only consider the case B= I in the main paper and present the general case as an extension in the appendix. P.S. Please put in clear that the strong convexity assumption is also respect to the metric B because we need it to pass from the last line of page 12 to line 345. 3) At line 56, a random variable \theta_k is introduced to obtain unbiased gradient. Is it true that such theta_k always exist for any sketching S_k? 4) At line 61, it is claimed that the variance of g_k goes to zero because h_k and g_k are estimate of the gradient. However, h_k is a biased estimate, which makes this statement non trivial. Please provide more details about it. 5) In the statement of Thm 3.3, we have no idea what \sigma is. Please make clear that it is in fact an intermediate parameter which is free to chose. 6) I suggest to put Thm E.1 into the main paper because it facilities the understanding of different parameters under the standard setting. Under the current presentation, we have no idea what is the order of the step-size alpha which turns out to be crucial for the convergence rate. 7) Section 4 is very confusing because we have no idea what the vector v is and the result seems to be overlapping with the previous section. 8) The experimental section is very unclear about what is the objective function. In particular, it is more interesting to compare with the standard coordinate descent method instead of projected gradient method, as shown in the appendix. It seems like coordinate descent always outperform the proposed algorithm, is there any intuition about this observation? EDIT: The author's rebuttal has carefully addressed my concerns. I believe it is a good paper worth to publication thanks to its theoretical contribution, which fills the gaps of non-separable proximal terms in the coordinate descent literature.

Reviewer 3



This paper presents a first order optimization method for composite optimization problems called SEGA (SkEtched GrAdient method). The method can be applied to problems where the smooth part of the of objective is strongly convex, the regularizer must be convex but not necessarily separable, and the proximal operator of the regularizer must also be available. The set up for this algorithm is that the user does not have access to the true gradient, but instead, an oracle returns a random linear transformation of the gradient. A random variable $\theta_k$ is introduced that is used to construct an unbiased estimate of the gradient. SEGA is a variance reduced algorithm because as iterations progress, the variance of the gradient estimate $g^k$ goes to $0$. The authors provide convergence theory for their algorithm, including proving a linear rate of convergence for SEGA under a smoothness and strong convexity assumption. A comparison of the complexity results for SEGA with complexity results for various coordinate descent methods is given in Table 1, and the results of several numerical experiments are also presented to demonstrate the practical performance of SEGA. The paper is well written and organized, although I encourage the authors to proof-read that paper carefully for typos (spelling and grammatical errors). The authors also took care to explain their work well, providing examples to help with understanding, and also referenced specific places in the appendices that supported claims made in the main body of the paper (this made it much easier to read/find).